# The decrease of intraflagellar transport impairs sensory perception and metabolism in ageing

Yincong Zhang[1,2,5], Xiaona Zhang [1,2,5], Yumin Dai[1,2], Mengjiao Song[1,2], Yifei Zhou [1,2], Jun Zhou [3,4], Xiumin Yan[1,2] & Yidong Shen [1,2✉]

Sensory perception and metabolic homeostasis are known to deteriorate with ageing, impairing the health of aged animals, while mechanisms underlying their deterioration remain poorly understood. The potential interplay between the declining sensory perception and the impaired metabolism during ageing is also barely explored. Here, we report that the intraflagellar transport (IFT) in the cilia of sensory neurons is impaired in the aged nematode *Caenorhabditis elegans* due to a *daf-19*/RFX-modulated decrease of IFT components. We find that the reduced IFT in sensory cilia thus impairs sensory perception with ageing. Moreover, we demonstrate that whereas the IFT-dependent decrease of sensory perception in aged worms has a mild impact on the insulin/IGF-1 signalling, it remarkably suppresses AMP-activated protein kinase (AMPK) signalling across tissues. We show that upregulating *daf-19*/RFX effectively enhances IFT, sensory perception, AMPK activity and autophagy, promoting metabolic homeostasis and longevity. Our study determines an ageing pathway causing IFT decay and sensory perception deterioration, which in turn disrupts metabolism and healthy ageing.

---

[1] State Key Laboratory of Cell Biology, Shanghai Institute of Biochemistry and Cell Biology, Center for Excellence in Molecular Cell Science, Chinese Academy of Sciences, Shanghai, China. [2] University of Chinese Academy of Sciences, Beijing, China. [3] Institute of Biomedical Sciences, College of Life Sciences, Key Laboratory of Animal Resistance Biology of Shandong Province, Collaborative Innovation Center of Cell Biology in Universities of Shandong, Shandong Normal University, Jinan, Shandong, China. [4] State Key Laboratory of Medicinal Chemical Biology, College of Life Sciences, Nankai University, Tianjin, China. [5]These authors contributed equally: Yincong Zhang, Xiaona Zhang. ✉email: yidong.shen@sibcb.ac.cn

Sensory perception is crucial to animal survival, not only for foraging and hazard avoidance but also for the regulation of metabolism[1]. Unfortunately, sensory perception declines with age, leading to one of the most common health problems in the aged population[2,3]. Despite its importance in health, the molecular mechanism of the ageing-induced deterioration of sensory perception remains poorly studied. The gradual loss of sensory neurons is considered as a major reason for the deteriorating sensory perception[4]. However, the nematode *Caenorhabditis elegans* (*C. elegans*), a well-established model organism for ageing research, exhibits olfactory deficits without losing any neurons[5,6]. In *C. elegans*, the neurocircuit of sensory perception starts from the cilia at the dendritic endings of sensory neurons, in a similar manner as olfactory perception in mammals[7–9]. A highly conserved intraflagellar transport (IFT) machinery, composed of motors and IFT complex proteins, delivers sensory receptors and other cargos bidirectionally along ciliary microtubules and is required for a functional cilium[10]. Despite the importance of sensory cilia in sensory perception, whether cilia degenerate with ageing and whether they cause the impairment of sensory perception in aged animals remain unexplored.

Metabolic homeostasis is critical in ageing. With ageing, the metabolic homeostasis is gradually disrupted, with catabolism (the breakdown of complex molecules to release energy) no longer matching with anabolism (the energy-consuming synthesis of complex molecules). Key catabolic pathways, such as autophagy, are dysregulated in aged animals[11,12]. Increasing catabolism or decreasing anabolism by modulating their pivotal regulators, TOR, AMP-activated protein kinase (AMPK) and insulin/IGF-1 signalling (IIS), was shown to effectively promote longevity by restoring metabolic homeostasis in various species[7]. Sensory perception is known to prime anabolism through activating pivotal anabolic pathways, such as IIS[1]. Consistently, mutating IFT genes extends the lifespan of the wild-type (WT) worms through *daf-16*/FOXO, a critical transcription factor inhibited by IIS[13,14]. However, loss of sensory perception also suppresses the longevity of the worms with mutated IGF receptor[14]. Besides, olfactory dysfunction is shown to be an early predictor of mortality in old age[4]. Therefore, sensory perception may play a complex role in ageing. It is intriguing to explore whether improving sensory perception can induce longevity and whether other pathways in addition to IIS are involved in the sensory-ageing regulation. In aged animals, the impact of the impaired sensory perception on the disruption of metabolic homeostasis is also unclear.

Here, we report a critical cause of the ageing-induced deterioration of sensory perception in *C. elegans*. IFT in the sensory cilia is disrupted with ageing because of a *daf-19*/RFX-dependent dysregulation of IFT protein expression, and in turn impairs sensory perception. Moreover, our results indicate that sensory cilia activate AMPK signalling autonomously in sensory neurons by *par-4*/LKB1 and non-autonomously in other tissues through the neurotransmitter octopamine. The ageing-induced deterioration of sensory perception thus contributes to the disruption of metabolic homeostasis. Upregulating *daf-19*/RFX improves IFT, sensory perception, AMPK activity and autophagy promoting both health span and lifespan. These findings not only highlight the *daf-19*/RFX-IFT axis in the degeneration of cilia and sensory perception with ageing, but also underscore the sensory perception-induced AMPK signalling as a critical factor in the age-related disruption of metabolic homeostasis.

## Results

**The intraflagellar transport in sensory cilia deteriorates with ageing**. To explore the effect of ageing on sensory perception, we first examined the response to food in young (day 1 of adulthood, D1) and aged worms (day 10 of adulthood, D10)[15]. Consistent with the age-related decline of chemotaxis[6,16], the food of bacteria no longer affected the movement of aged worms whereas young adults exhibited a clear enhanced slowing response (ESR) to bacteria (Fig. 1a, Supplementary Fig. 1a), indicating a defect of sensory perception with ageing. Cilia defects suppress dye-filling in sensory neurons[17]. Subsequent dye-filling assay indicated that the staining of DiI in the soma of sensory neurons became remarkably weaker at D10 (Fig. 1b), suggesting that ageing causes defects in the sensory cilia. Following this clue, we next examined age-related changes in cilia.

We first examined the cilia length in worms at D1 and D10 (Supplementary Fig. 1b) and found no obvious changes, implying that the ciliary structure is unaffected by ageing at D10. Consistently, it has been shown that the microtubule (MT) organisation in sensory cilia does not suffer any obvious changes in aged worms[5]. In addition to the MT bundles, molecular trafficking along MT (i.e., IFT) is also crucial to cilia function[10]. We then examined IFT by live imaging in young and aged worms. The anterograde IFT is driven by kinesin and IFT-B complex, whereas the retrograde IFT by dynein and IFT-A complex (Fig. 1c). The motors of kinesin (OSM-3/KIF17), dynein (CHE-3/DHC2) and the core components of IFT-B (OSM-6/IFT52) and IFT-A (CHE-11/IFT140) complexes were endogenously GFP-tagged using genomic editing to visualise IFT in vivo. The length of GFP signal along cilia from CHE-3::GFP and CHE-11::GFP exhibited a slight increase in aged worms (Supplementary Fig. 1c and 1d), implying that IFT may be dysregulated in aged worms[18,19]. Indeed, both the frequencies and velocities of these IFT components were remarkably decreased in the sensory cilia of aged worms (Fig. 1d–f and Supplementary Movie 1). As the rate of ageing is highly variable among individuals[20], the decrease of IFT also exhibited a wide variation in aged worms (Supplementary Fig. 1e). *daf-2(-)* is a well-established longevity mutant with defective IIS[7]. Whereas IFT was reduced in D10 WT worms (Fig. 1), it was well-maintained in *daf-2(-)* mutants at D10 and D20 (Supplementary Fig. 2a–c and Supplementary Movie 2). Therefore, IFT decreases with ageing whereas is protected in the longevity mutant of *daf-2(-)*.

**DAF-19 is critical in the ageing-induced decline of IFT and sensory perception**. The proper assembly of the multi-component IFT complexes is required for IFT. Inhibiting the expression of IFT complex components is known to suppress IFT[10,21]. Therefore, to explore the mechanism underlying the decreasing IFT with ageing, we examined the expression of IFT components in WT worms at D1 and D10. In the soma of amphid and phasmid sensory neurons, endogenously GFP-tagged OSM-3 and OSM-6, but not CHE-11 or CHE-3, exhibited a remarkable decrease with ageing in the WT worms. In phasmid sensory cilia, all four examined IFT components were reduced during ageing (Fig. 2a). DAF-19, an RFX transcription factor, is a master regulator of IFT genes in *C. elegans*[22]. Consistent with the decrease of IFT components with ageing, endogenously GFP-tagged DAF-19 was downregulated in the neurons of aged WT worms (Fig. 2b), implying that *daf-19* is critical in maintaining IFT during ageing.

To investigate the role of *daf-19* in IFT, we knocked down *daf-19* in neurons. GFP in sensory neurons were significantly suppressed by neuron-specific RNAi against GFP (Supplementary Fig. 3a). A set of *daf-19*-regulated IFT genes expressed in sensory neurons were remarkably suppressed upon neuron-specific *daf-19* RNAi, as well (Supplementary Fig. 3b). Therefore, *daf-19* RNAi were effective in sensory neurons. Meanwhile, the inhibitory

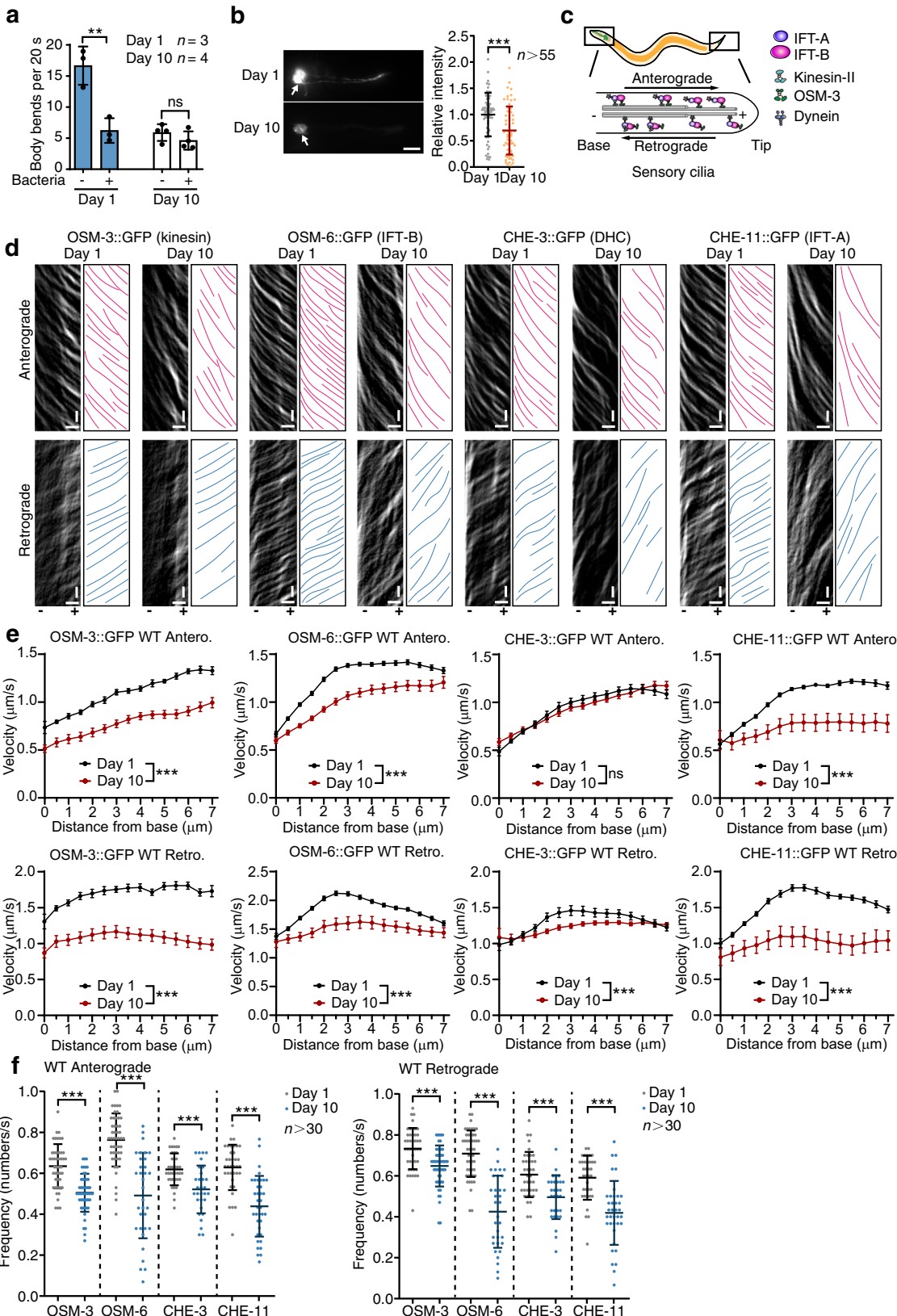

effect of *daf-19* RNAi was milder than *daf-19* null mutants because it did not abolish ciliogenesis[14] (Supplementary Fig. 3c). As we hypothesised, *daf-19* RNAi substantially downregulated the motilities of examined IFT components in WT worms and *daf-2* (-) mutants (Supplementary Fig. 3c–e). Overexpressing *daf-19c*, a *daf-19* isoform which specifically regulates ciliary genes, with its native promoter effectively ameliorated the decrease of DAF-19 in aged WT worms[22,23] (Fig. 2c). As a result, overexpressing *daf-19c* increased all four examined IFT proteins in the amphid/phasmid neurons soma of aged worms (Fig. 2d). In young worms, *daf-19c* overexpression still cause a robust increase of the examined IFT components in the soma of amphid neurons and CHE-3::GFP in

**Fig. 1 The intraflagellar transport (IFT) in the sensory cilia is decreased by ageing. a** The enhanced slowing response to the food of worms at indicated ages. Sensing the food reduces worm movement (body bends). The worms at day 10 of adulthood are over the reproductive age and exhibit ageing features. $n = 3$ (day 1) or $n = 4$ (day 10) biological independent experiments. $p = 0.0078$ and 0.2493, respectively. **b**. The DiI staining is decreased in the sensory neurons of aged worms ($n = 66$ and 62 animals respectively. $p = 0.0001$.). Arrows denote the soma of sensory neurons. Scale bar: 10 μm. **c**. A depiction of IFT in sensory cilia, which are at the dendritic endings of sensory neurons. IFT-B complex and kinesin control the anterograde IFT, whereas IFT-A complex and dynein modulate the retrograde IFT. **d–f** IFT decreases with ageing. Representative IFT components were examined at indicated ages. The representative kymographs (**d**), velocities (**e**) and frequencies (**f**) are respectively shown. For **d**, similar results were obtained in all independent experiments. Exact sample size and $p$ value are included in Source Data file. (+) and (−) denote microtubule polarity. Horizontal scale bars: 2 μm; vertical scale bars: 2 s. Data are presented as mean ± SEM in **e** and mean ± SD in the rest. Two-way ANOVA in **e**, and unpaired $t$-test (two-tailed) in the rest, $**p < 0.01$, $***p < 0.001$, ns non-significant. Source data are provided as a Source Data file.

the soma of phasmid neurons (Fig. 2d). Following the increase of IFT proteins, both the velocity and frequency of the four examined IFT components were improved at D10 when *daf-19c* was upregulated. In young worms at D1, overexpressing *daf-19c* had a weaker effect, but still increased both anterograde and retrograde IFT (Fig. 3a–c and Supplementary Movie 3). Consistently, overexpressing *daf-19c* suppressed the diminishing DiI staining in sensory neurons with ageing (Fig. 3d). Therefore, a DAF-19-regulated decrease of IFT components underlies the age-related decline of IFT and in turn impairs the function of sensory cilia in aged worms.

We next performed chemotaxis assay to check whether the DAF-19-IFT-sensory cilia axis is also responsible for the deteriorating sensory perception with ageing. Because the worms at D10 suffer a severe decrease of motility and are not suitable for chemotaxis assay, worms at D5, which are at the end of their reproductive period, and young worms at D1 were examined. As expected, overexpressing *daf-19c* abolished the decreased attraction to butanone and repulsion to nonanone at D5 (Fig. 3e and Supplementary Movie 4). Consistently, the response to food at D10 was also improved by upregulating *daf-19c* (Fig. 3f). Taken together, these results indicate that overexpressing *daf-19c* effectively suppresses the ageing-induced degeneration of sensory cilia and perception.

**Food perception through sensory cilia activates AMPK signalling.** Sensory perception, which requires proper IFT in sensory cilia, is tightly related to metabolism[1,10]. Since *daf-19c* controlled IFT underlies the age-related decline of sensory perception, we next examined its impact on metabolism. Metabolism is composed of the biosynthetic anabolism and the energy-yielding catabolism. In *C. elegans*, sensory cilia are known to modulate insulin/IGF-1 signalling (IIS), a critical signalling pathway in anabolism[3,14]. IFT mutants of *osm-3(-)* and *osm-6(-)* exhibited severe cilia defects and IIS target genes were upregulated as reported at D1[13,14] (Supplementary Fig. 4a). In aged worms, IIS exhibited a decreased modulation by sensory perception, as multiple IIS target genes were no longer changed upon mutating *osm-3* or *osm-6* at D10 (Supplementary Fig. 4a). Overexpressing *daf-19c* by its own promoter increases the mRNA level of *daf-19c* and its target genes (Supplementary Fig. 4b), whereas failed to change IIS target genes expression in either young or aged worms (Supplementary Fig. 4c), implying that the enhanced sensory perception may not interfere with IIS.

We next examined the effect of sensory perception on AMPK, a pivotal driver of catabolism[24]. We first used western blot to check the level of activated AMPK (phosphorylated at the conserved Thr172, p-AMPK) in the whole worm. Overexpressing *daf-19c* in sensory neurons with its native promoter or in pan neurons with a neuron-specific promoter remarkably increased p-AMPK levels in young worms[22,23] (Fig. 4a, b), whereas the fluorescent markers of these strains have no effect (Supplementary Fig. 5a). Due to an increasing ratio of AMP versus ATP[25], p-

AMPK is increased in aged worms (Fig. 4b). Overexpressing *daf-19c* further upregulates p-AMPK in worms at D10 (Fig. 4b), indicating its robust control on AMPK throughout ageing. Consistently, downregulating *daf-19* in neurons decreases p-AMPK in both young and old worms (Fig. 4c). To examine whether this could be due to any side effects in larval development, we prepared another strain to overexpress *daf-19c::degron::gfp* by its native promoter. The overexpression of *daf-19c::degron::gfp* was suppressed by auxin treatment during larval stages and induced specifically in adulthood by removing auxin[26] (Supplementary Fig. 5b, c). Whereas inhibiting *daf-19c* upregulation in larvae effectively blocked the increase of p-AMPK in this strain at D1, the adulthood specific overexpression of *daf-19c* still enhanced p-AMPK level at D10 (Supplementary Fig. 5d). Therefore, it is unlikely that the *daf-19c*-induced p-AMPK is due to secondary effects in larval development. As western blot shows p-AMPK levels from the whole body, these results also imply that *daf-19c* could non-autonomously induce AMPK signalling in other tissues. Activated AMPK phosphorylates the CREB regulated transcriptional coactivator (CRTC-1) in *C. elegans* and induces its cytosolic translocation[27]. For further confirmation, we then examined the nuclear localisation of CRTC-1::RFP in the intestine. As expected, overexpressing *daf-19c* reduced the nuclear localisation of CRTC-1::RFP in the intestine (Fig. 4d), confirming that *daf-19c* in sensory neurons activates AMPK in other tissues.

To further test whether the *daf-19c*-induced AMPK activity is due to the enhanced sensory perception or other effects by *daf-19c* upregulation, worms were incubated in the plates with food, with food odour (food on the lid) or without food and examined for p-AMPK levels. The *daf-19c*-induced AMPK activation occurred only when worms sensed the food or food odour (Fig. 4a), indicating that it does require sensory perception, especially olfactory perception of food. Moreover, disrupting IFT and sensory cilia via mutating *osm-3* fully abolished the elevated p-AMPK levels in both young and aged worms overexpressing *daf-19c* (Fig. 4e and Supplementary Fig. 5e). Therefore, *daf-19c* activates AMPK by enhancing IFT in sensory cilia and improving food perception. Consistently, when sensory cilia are disrupted by the mutation of *osm-3(-)* or *osm-6(-)*, p-AMPK levels decreased at D1 and D10 (Fig. 4e, f), confirming that sensory perception promotes AMPK activity. As *daf-19* controls innate immunity[28], worms were further incubated on UV-killed bacteria to minimise immunity response and examined for p-AMPK levels. *daf-19c* overexpression still robustly increased p-AMPK in worms fed with UV-killed bacteria (Supplementary Fig. 5f), indicating that this effect is independent of innate immunity.

Primary cilia in cultured cells modulate AMPK via LKB1, a kinase phosphorylating AMPK[29]. By tagging the worm ortholog of LKB1 (PAR-4) with mNeonGreen using genomic editing, we found that it was expressed in sensory neurons (Fig. 5a). Using a neuron-specific transgene with a higher *par-4* expression, we further observed PAR-4::GFP in cilia (Fig. 5b), as reported for

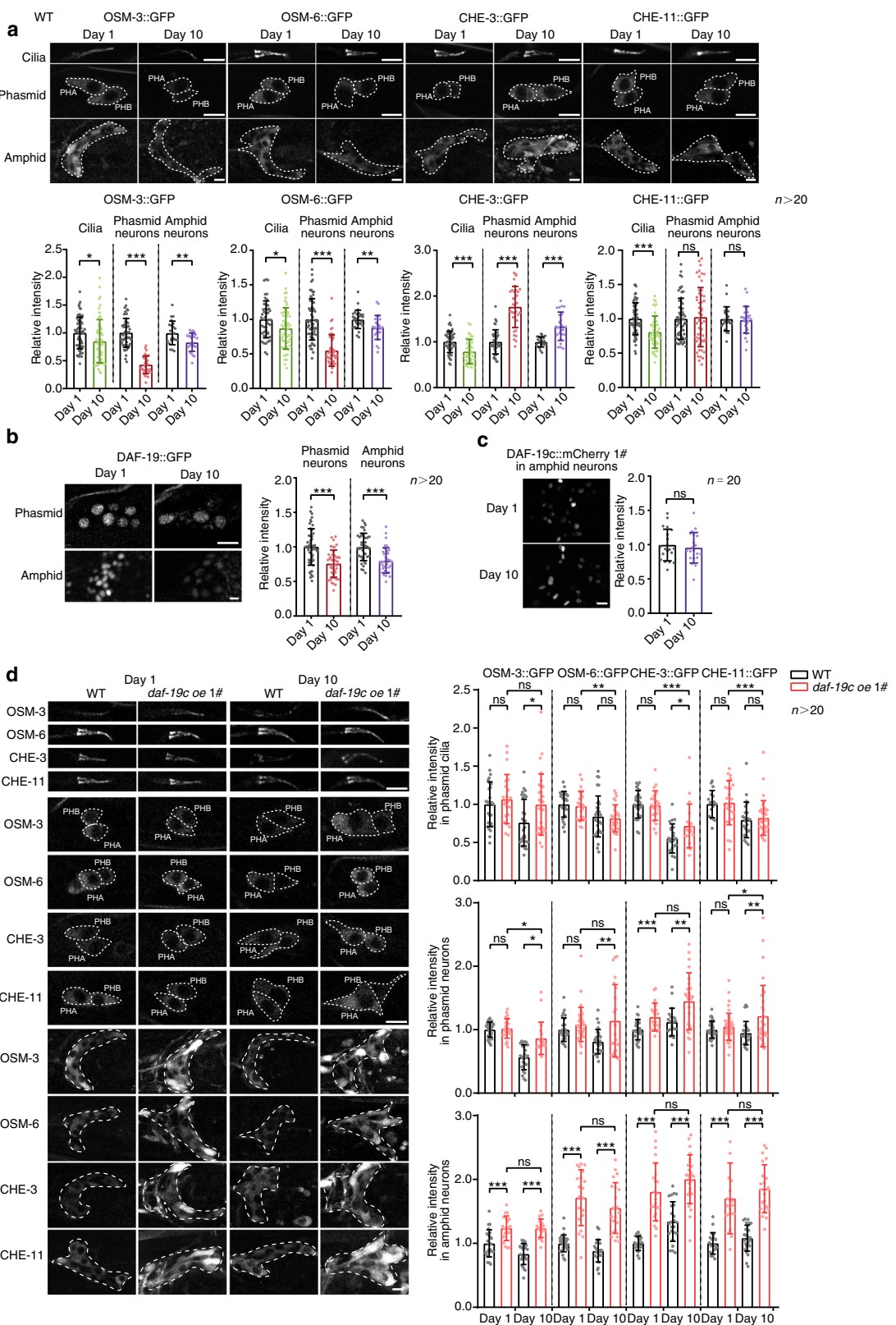

LKB1 in mammalian cilia[29,30]. As *daf-19c* promotes IFT, overexpressing *daf-19c* consistently increased PAR-4::GFP localisation in cilia (Fig. 5b). RNAi against *par-4* specifically in all neurons or in sensory neurons blocked the increase of p-AMPK in worms overexpressing *daf-19c* (Fig. 5c and Supplementary Fig. 6a), indicating that sensory cilia control AMPK activity via *par-4*. FLCN is required for LKB1 localisation to cilia[31]. Similarly,

inhibiting the worm ortholog of FLCN, *flcn-1*, suppressed the enhanced ciliary localisation of PAR-4::GFP and increased p-AMPK in the worms overexpressing *daf-19c* (Supplementary Fig. 6b–c), suggesting that *daf-19c* regulates AMPK activity through a *flcn-1-par-4* axis.

We next pursued the molecule transducing AMPK signalling non-autonomously from sensory neuron. Activated AMPK in

**Fig. 2 IFT components are downregulated with ageing in a DAF-19/RFX-dependent manner. a** The expression of the indicated IFT components in the phasmid cilia and the soma of phasmid and amphid neurons of WT worms at indicated ages. The tested proteins were endogenously tagged with GFP using CRISPR/Cas9 technology. Scale bars: 5 μm. $n > 20$ animals in each experiment. Exact sample size and $p$ value are included in Source Data file. **b** The expression of endogenously tagged DAF-19::GFP in the soma of phasmid and amphid neurons of WT worms at indicated ages. Scale bars: 5 μm. $n > 20$ animals in each experiment. Exact sample size and $p$ value are included in Source Data file. **c** The expression of DAF-19c::mCherry in the soma of amphid neurons at indicated ages. Scale bar: 5 μm. $n = 20$ animals in each experiment. $p = 0.5763$. **d** The expression of IFT components in the phasmid cilia, phasmid neurons and amphid neurons in the indicated strains at day 1 and day 10 of adulthood. The worms overexpressing *daf-19c* were examined in parallel to the WT worms in A. Scale bar: 5 μm. $n > 20$ animals in each experiment. Exact sample size and $p$ value are included in Source Data file. Data are presented as mean ± SD. Unpaired $t$-test (two-tailed), $*p < 0.05$, $**p < 0.01$, $***p < 0.001$, ns non-significant. Source data are provided as a Source Data file.

neurons affects other tissues through octopamine[32]. To examine whether octopamine is required in the sensory perception-induced AMPK signalling, two octopamine biosynthetic enzymes (*tbh-1* and *tdc-1*) were mutated[32]. Indeed, mutating either of them blocked the increase of p-AMPK in worms overexpressing *daf-19c* (Fig. 5d, e). Supplementing the mutants of *tbh-1(-)* and *tdc-1(-)* with octopamine rescued the elevated p-AMPK induced by *daf-19c* overexpression (Fig. 5e). Therefore, octopamine is involved in the sensory perception-induced AMPK signalling.

**Sensory perception promotes metabolic homeostasis and longevity.** Activated AMPK is a positive regulator of longevity in *C. elegans*, driving critical catabolic processes including autophagy[7,25]. Since enhanced sensory perception activates AMPK (Figs. 4 and 5), we next explored whether it also promotes autophagy in the intestine using a mCherry-GFP-tagged LGG-1 reporter. GFP in this reporter is specifically quenched in auto-lysosomes (ALs), thus labelling autophagosomes (APs) with both mCherry and GFP and ALs with mCherry[12]. As expected, upregulating *daf-19c* in sensory neurons remarkably increased the number of ALs whereas mildly reduced APs in the intestine (Fig. 6a). Either blocking or enhancing autophagy flux could change the numbers of APs and ALs. Chloroquine blocks autophagy flux and should not regulate APs and ALs when autophagy is already blocked[12]. Chloroquine suppressed the change of APs and ALs upon *daf-19c* overexpression (Fig. 6a), indicating that autophagy is active in the worms overexpressing *daf-19c* and enhanced sensory perception promotes autophagy in the intestine.

Metabolism is closely related to ageing. We next examined a series of hallmarks of healthy ageing in worms overexpressing *daf-19c*. Overexpressing *daf-19c* improved chemotaxis, the enhanced slowing response (ESR) to food and motility in aged worms (Figs. 3e, f and 6b, and Supplementary Fig. 7a), which are key health metrics[15]. Interfering AMPK activity by neuron-specific RNAi against either *par-4*/LKB1 or *aak-2*/AMPK abrogated the enhanced motility in the worms overexpressing *daf-19c* at D10 (Supplementary Fig. 7b), indicating that sensory perception promotes healthy ageing through AMPK signalling. The motility of worms is closely related to the integrity of myofibers in the body wall muscle (BWM), which is prone to ageing[20,33]. Sensory neuron-specific overexpression of *daf-19c* reduced myofilament abnormalities in aged worms, whereas a neuron-specific RNAi against *daf-19* in adult worms had an adverse effect (Supplementary Fig. 7c, d). AMPK and autophagy are critical in maintaining the balance of protein metabolism and in turn promote health and longevity[7]. The accumulation of polyglutamine (polyQ) aggregates is a marker of the deteriorating protein homeostasis[12]. Consistently, overexpressing *daf-19c* reduced polyQ aggregates (Fig. 6c), indicating improved protein homeostasis with the enhanced sensory perception. Taken together, sensory perception promotes healthy ageing through AMPK.

We next checked the potential effect of sensory perception on lifespan. With ageing, the motilities of IFT components decrease at variable rates among worms (Supplementary Fig. 1e). We then examined the lifespan of worms with remarkably different velocities of IFT components (i.e., OSM-3, OSM-6 and CHE-11) at D10. The total population of examined worms had a similar lifespan as the untreated WT worms in the other assays (Supplementary Fig. 7e; Source Data), indicating that live imaging had little effect on the ageing assay. In this assay, the worms with faster IFT lived longer than the worms with slower IFT (Fig. 6d and Supplementary Fig. 7e; Source Data), showing a positive correlation between IFT function and longevity. We further examined the lifespan of the worms with improved sensory perception. Indeed, two strains overexpressing *daf-19c* with its native promoter, and another strain overexpressing *daf-19c* with a neuron-specific promoter all exhibited extended lifespans (Fig. 6e and Supplementary Fig. 7f; Source Data). Disrupting sensory cilia by an adult-specific RNAi against *osm-3* abrogated the extended lifespan of the worms overexpressing *daf-19c* (Fig. 6f; Source Data), indicating that *daf-19c* promotes longevity by enhancing sensory perception. Since sensory perception activates AMPK to improve worm motility (Supplementary Fig. 7b), neuron-specific RNAi against *par-4*/LKB1 or *aak-2*/AMPK was performed in adult worms to test whether the sensory perception-induced longevity also requires AMPK. Indeed, the lifespan of worms overexpressing *daf-19c* is reduced to the same level as the WT worms upon either of the two RNAi treatments (Fig. 6g; Source Data). Therefore, sensory perception promotes longevity via upregulating AMPK signalling.

## Discussion

The decline of sensory perception is a hallmark of ageing[2]. However, its underlying mechanism remains poorly understood. Here, we report in the nematode *C. elegans* that the dysfunction of the sensory cilia, the start point of sensory circuit, is a major cause of the ageing-induced decrease of sensory perception. With ageing, a reduction of the master transcription factor of IFT genes, DAF-19/RFX, downregulates IFT components, disrupts the multiprotein IFT complexes, decreases IFT in the sensory cilium, and thereby impairs sensory perception. Upregulating *daf-19c* is an effective way to suppress these changes by rescuing the expression of IFT components in aged worms (Fig. 6h). Cilia are important regulators of cell survival and functions[34]. The dysfunction of cilia is known to impair neuronal activity[35,36]. Meanwhile, it is reported that brain injury induces cilia defects[36]. It will be interesting to pursue in future the intertwined inter-action between cilia and neuronal activity. As the dysregulation of neurotransmitters and the loss of sensory neurons are involved in the age-related decrease of sensory perception, whether the degeneration of sensory cilia primes these changes is also an intriguing topic to explore[16,37,38].

The four IFT components we assayed are affected at different degrees by ageing and *daf-19* (Fig. 1). IFT trains are multiple protein complex assembled in a stepwise manner from its core,

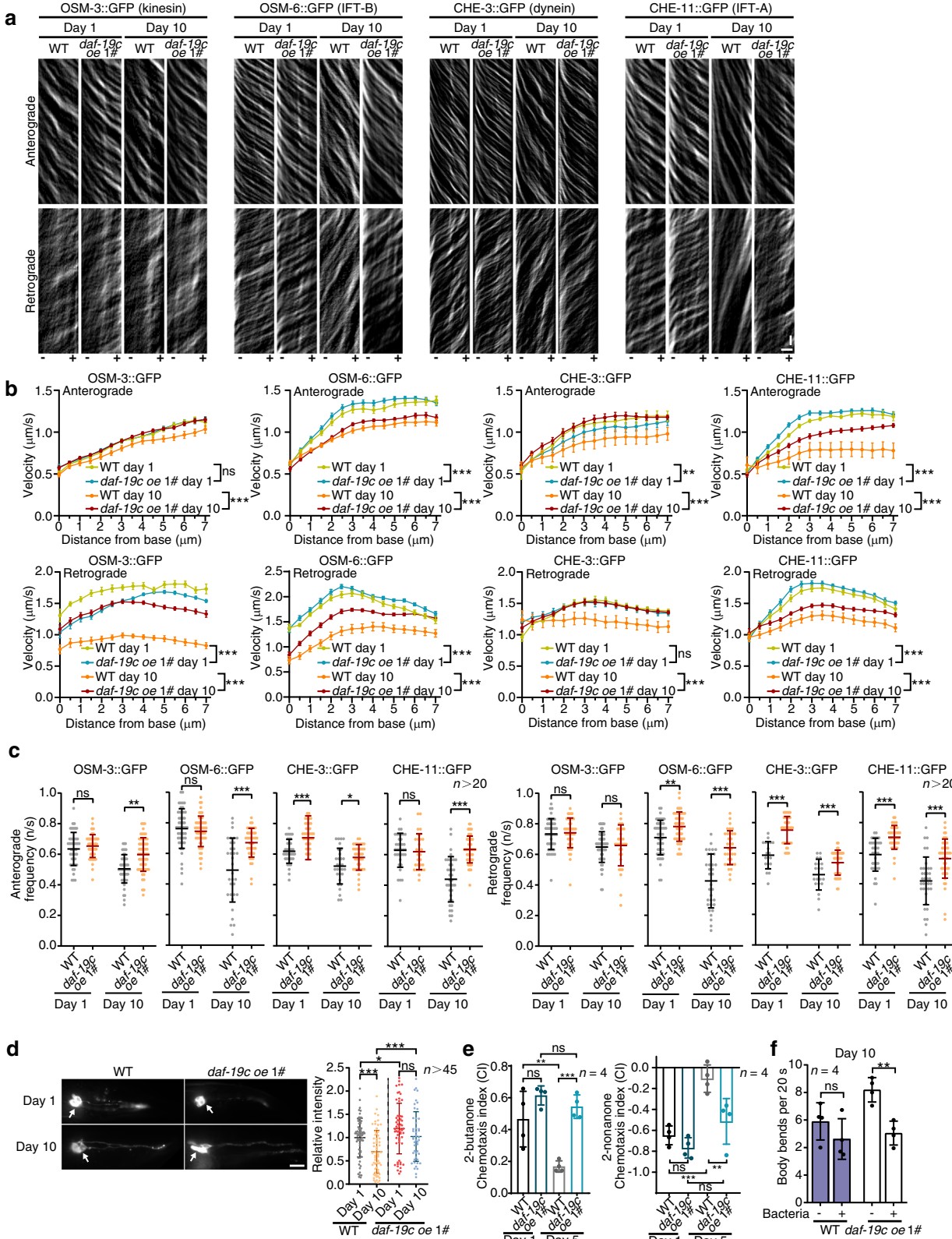

the motors[39]. In vitro reconstruction of worm IFT complex indicates that IFT trains without critical components can still move but at a lower speed, implying that IFT complex with various compositions could exhibit different motilities. As ageing downregulates IFT components differently (Fig. 2), the compositions may vary from one IFT complex to another in aged worms, making IFT trains labelled by different components suffer

different impacts from ageing. Similarly, as *daf-19* drives IFT genes at various levels, its regulation on the motility of different IFT components also varies (Fig. 3). But due to the technical limits, we failed to distinguish the individual IFT trains with different motilities and compositions in our live imaging assays. With the development of super resolution microscopy, it will be intriguing to pursue this issue in the future.

**Fig. 3 Overexpressing *daf-19*/RFX enhances IFT and sensory perception in the aged worms. a–c** Overexpressing *daf-19c* with its native promoter increases IFT in the sensory cilia at indicated ages. Representative kymographs, velocities and frequencies are respectively shown in **a–c**. For **a**, similar results were obtained in all independent experiments. Horizontal scale bars: 2 µm; vertical scale bars: 2 s. Exact sample size and *p* value are included in Source Data file. **d** Overexpressing *daf-19c* rescues the impaired DiI staining in the aged sensory neurons. Arrows denote the soma. The assay was performed in parallel with Fig. 1b. Scale bar: 10 µm. *n* = 66, 62, 59, 47, respectively. *p* = 0.0001(WT day 10 versus WT day 1), 0.026 (*daf-19c oe* day 1 versus WT day 1), 0.1092 (*daf-19c oe* day 10 versus *daf-19c oe* day 1) and 0.0008 (*daf-19c oe* day 10 versus WT day 10). **e** Attraction (2-butanone) and repulsion (2-nonanone) chemotaxis assays of indicated strains at day 1 and day 5 of adulthood. *n* = 4 biological independent experiments. Exact sample size and *p* value are included in Source Data file. **f** *daf-19c* overexpression improves the response to food in the aged worms. Note that a decrease in body bends upon feeding indicates the response to food. *n* = 4 biological independent experiments. Exact sample size and *p* value are included in Source Data file. Data are presented as mean ± SEM in **b** and mean ± SD in the rest. Two-way ANOVA in **b**, one-way ANOVA in **e** and unpaired *t*-test (two-tailed) in the rest, *$p < 0.05$, **$p < 0.01$, ***$p < 0.001$, ns non-significant. Source data with exact *p* values are provided as a Source Data file.

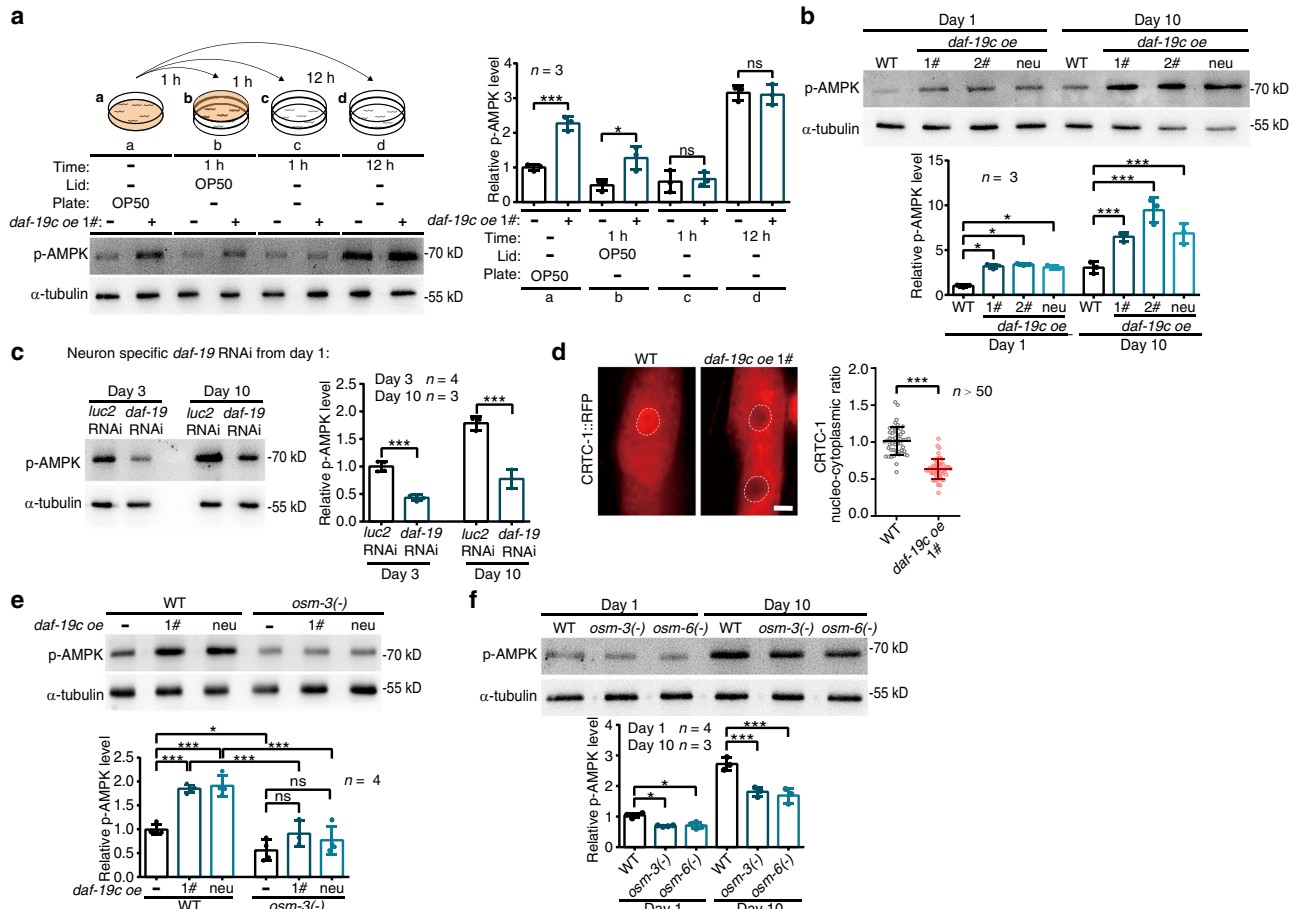

**Fig. 4 The sensory cilia induce AMPK activity across tissues. a** The phosphorylation levels of AMPK (p-AMPK) in young WT worms subjected to indicated treatments. The bacteria of OP50 (orange) is the food of *C. elegans*. Note that *daf-19* upregulates p-AMPK only when worms are fed (OP50 in the plate, **a**) or smell the food (OP50 on the lid, **b**). *n* = 3 biological independent experiments. Exact *p* value are included in Source Data file. **b** Overexpressing *daf-19c* by its native promoter (1# and 2#) or a neuron-specific promoter (neu) increases p-AMPK levels in both young and aged worms. *n* = 3 biological independent experiments. Exact *p* value are included in Source Data file. **c** Downregulating *daf-19* in neurons from adulthood reduces p-AMPK levels in both young and aged worms. *n* = 3 biological independent experiments. Exact *p* value are included in Source Data file. **d** Overexpressing *daf-19c* reduces the nuclear localisation of CRTC-1::RFP in the intestinal cells. Dotted lines denote the nucleus of intestinal cells. Scale bar: 10 µm. *n* = 51 and 52 animals, respectively. *p* < 0.0001. **e** Disrupting cilia by mutating *osm-3* suppresses the increase of p-AMPK in the worms overexpressing *daf-19c* at day 1 of adulthood by its native promoter (1#) or a neuron-specific promoter (neu). *n* = 3 biological independent experiments. Exact *p* value are included in Source Data file. **f** p-AMPK levels decrease in worm mutants with defected cilia (*osm-3(-)*, *osm-6(-)*). *n* = 3 biological independent experiments. Exact *p* value are included in Source Data file. α-tubulin serves as loading controls in western blot assays. Data are presented as mean ± SD. Unpaired *t*-test (two-tailed) in **d**, and one-way ANOVA in the rest. *$p < 0.05$, ***$p < 0.001$, ns non-significant. Source data with exact *p* values are provided as a Source Data file.

Metabolic homeostasis relies on delicate modulations on critical regulators of metabolisms. Food perception without ingestion is known to drive insulin/IGF-1 signalling (IIS)[1]. Our results indicate that it also directly induces AMPK signalling (Fig. 6h). Consistently, autophagy, an essential catabolic progress under AMPK regulation, is also enhanced by improving sensory perception (Fig. 6h). Therefore, food perception simultaneously controls the pivotal regulators of both anabolism (IIS) and catabolism (AMPK). The two pathways counteract with each other to maintain the balance of metabolism in response to the upcoming influx of nutrients[40].

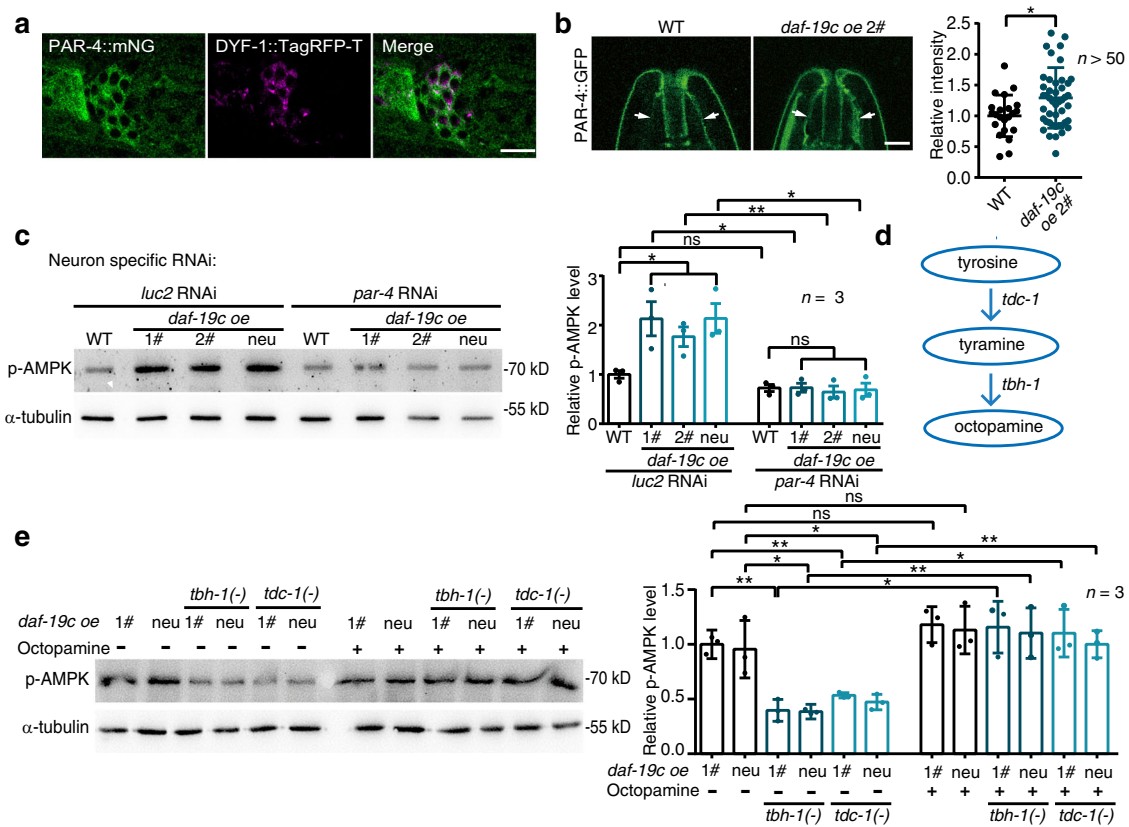

**Fig. 5 The enhanced sensory perception upregulates AMPK activity through *par-4* and octopamine. a** PAR-4 (green) is expressed in neurons, including sensory neurons. Similar results were obtained in all independent experiments. The endogenous *par-4* gene was tagged with mNeonGreen (mNG) using CRISPR/Cas9 technology. DYF-1::TagRFP-T (magenta) marks sensory neurons. Scale bar: 10 μm. **b** The ciliary localisation of PAR-4::GFP (arrows) is enhanced by overexpressing *daf-19c*. Scale bar: 5 μm. $n = 66$ and 58 animals, respectively. $p = 0.0233$. **c** The neuron-specific RNAi against *par-4* blocks the upregulation of p-AMPK in worms overexpressing *daf-19c*. $n = 3$ biological independent experiments. Exact $p$ value are included in Source Data file. **d** *tdc-1* and *tbh-1* are two genes encoding critical enzymes in octopamine synthesis. **e** The levels of p-AMPK in the indicated worms. Worms were collected at day 1 of adulthood post 30 min of 4 mM octopamine or mock treatment. $n = 3$ biological independent experiments. Exact $p$ values are included in Source Data file. α-tubulin serves as loading controls in western blot assays. Data are presented as mean ± SEM in **c** and mean ± SD in the rest. Unpaired $t$-test (two-tailed) in **b**, **c**, one-way ANOVA in **e**, \*$p < 0.05$, \*\*$p < 0.01$, \*\*\*$p < 0.001$, ns non-significant. Source data are provided as a Source Data file.

Improving sensory perception upregulates AMPK activity but with little effect on IIS. This suggests different regulation thresholds of catabolism and anabolism by sensory perception and implies a metabolic protection against obesity when worms smell overabundant food. Consistently, increasing olfactory sensitivity suppresses diet-induced obesity in mice[41]. Besides, sensory defect has a diminishing control on IIS in the aged worms, whereas robustly controls AMPK throughout ageing (Fig. 4 and Supplementary Fig. 4). This implies different temporal controls on these two pathways from sensory perception and that the modulation of sensory perception on metabolism shifts to AMPK in the aged worms.

The dual control on IIS and AMPK by sensory perception thus makes it possible to promote worm longevity by modulating sensory in both directions. Whereas sensory defect from birth extends lifespan by suppressing IIS[13,14], improving sensory promotes longevity by enhancing AMPK (Fig. 6h). The impact of IIS on longevity is much wider than AMPK signalling[7,42]. Sensory mutants, which are defective in AMPK activation (Fig. 4), still live longer due to the overwhelming pro-longevity effect from the suppressed IIS[14]. Consistently, multiple reports indicate that IIS mutants with reduced AMPK are longer lived than WT worms or AMPK-defective mutants[43–45]. Since AMPK is only one of the downstream effectors of IIS, defective sensory perception reduces AMPK (Fig. 4), and thus suppresses the longevity of *daf-2(-)* mutants[14].

As the control of sensory perception on IIS weakens during ageing, we speculate that the deterioration of sensory perception in aged worms could impact the metabolic homeostasis mainly by suppressing AMPK. Therefore, sensory perception improvement is an effective way to maintain metabolic homeostasis in aged worms by elevating catabolism, and in turn promotes healthy ageing and longevity. Similarly, disrupting sensory perception impairs lipid catabolism and WT worms with better sensory perception live longer[16,46] (Fig. 6).

The ciliary function could also decline in aged people. Besides, loss of sensory perception is a risk factor to death and linked to many age-related diseases, such as obesity and neurodegenerative disease[1,4], suggesting that cilia could modulate ageing in human. As all the genes in this study are evolutionarily conserved, it will be interesting to see if similar pathways underlie age-related diseases and the metabolic changes in aged vertebrates.

## Methods

***C. elegans* strains and culture**. *C. elegans* strains used in this study are listed in Supplementary Data 1. Worms were grown on NGM plates with standard techniques at 20 °C[47]. All assayed worms were at day 1 of adulthood unless otherwise noted. Some strains were provided by CGC, which is funded by NIH Office of Research Infrastructure Programs (P40 OD010440). OP50 colonies on an NGM plate were treated with the sterilisation programme in a GS Gene Linker UV Chamber (BIO-RAD) for 15 m to obtain UV-killed bacteria. The killed bacteria were confirmed by no growth after O/N incubation in LB.

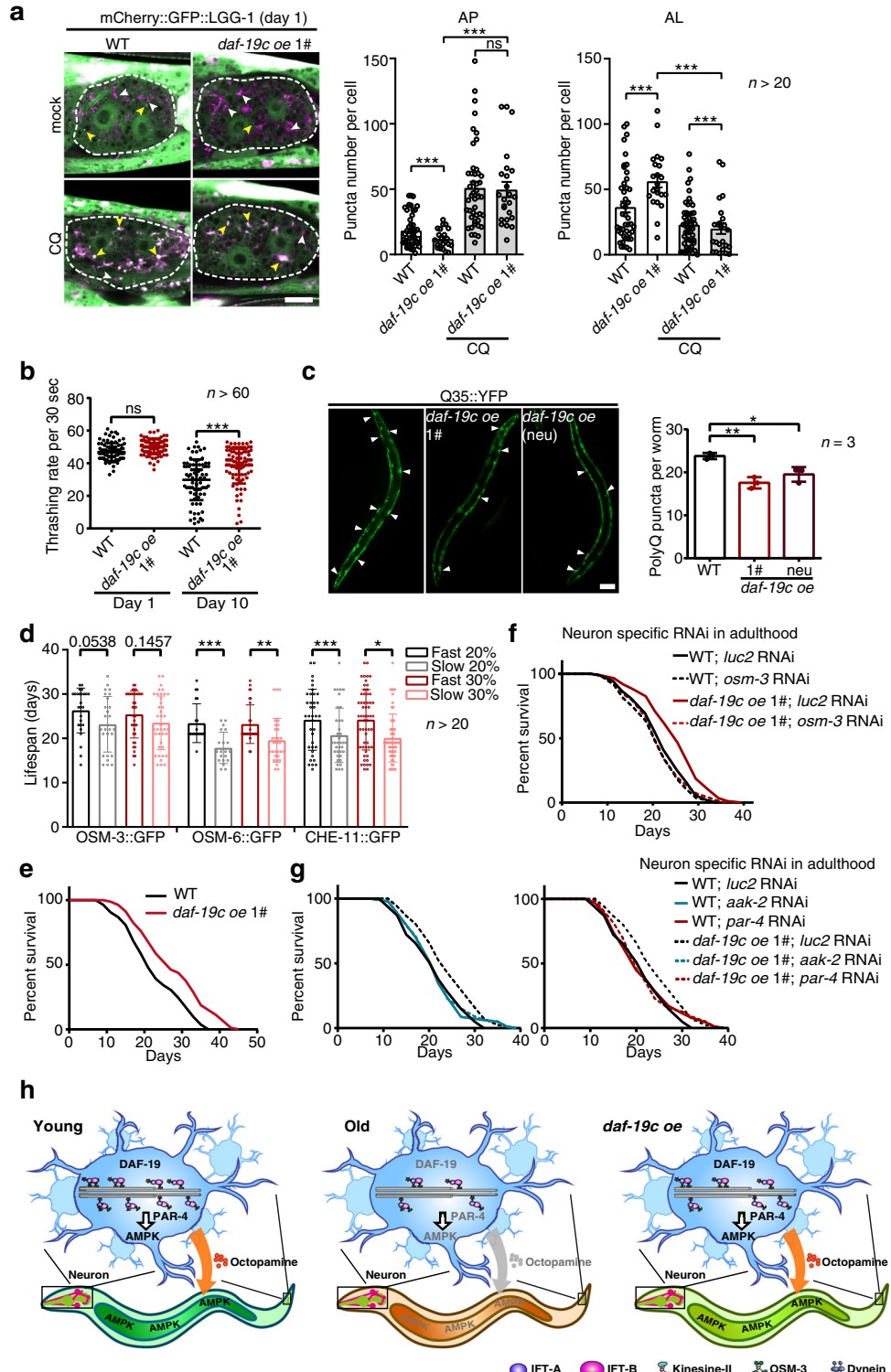

We have complied with all relevant ethical regulations for animal testing and research. The study has been approved by the Ethics Committee of Shanghai Institute of Biochemistry and Cell Biology, CAS.

**Lifespan assays**. All lifespan assays were performed at 20 °C. For synchronisation, L4 worms from eggs laid in a time window of 4 h or O/N were picked. Worms were transferred away from progeny to fresh plates every other day during the reproductive period. Worm survival was scored every 2 or 3 days. Worms undergoing internal hatching, bursting vulva or crawling off the plates were censored. Worms not responding to prodding were scored as dead. Graphpad Prism (GraphPad Software) was used to plot survival curves and calculate median lifespan. Statistical analysis was performed with the Mantel-Cox Log Rank method.

For lifespan assays with correlated IFT velocities, individual N2 worms at day 10 of adulthood were anaesthetised using 5 mM levamisole and examined for IFT velocity on agar pads by live imaging microscopy. After imaging, worms were immediately recovered with M9 buffer and individually incubated in 35 mm NGM plates for ageing assay. At least 100 worms were tested in total.

**Plasmid construction**. All plasmids used in this study were constructed by Gibson Assembly. Primers used in plasmid constructions can be found in Supplementary Data 2. To generate *L3781-Pdaf-19c::daf-19c::mCherry*, 2321 bp of the promoter and coding sequence of *daf-19c* were PCR amplified from N2 genomic DNA and cloned into *L3781-mCherry*[22]. To generate *L3781-Pdaf-19c::daf-19c::degron::gfp*, *degron::gfp* was amplified and insert into *L3781-Pdaf-19c::daf-19c*. *degron::gfp* was a

**Fig. 6 Improving sensory perception promotes the health and survival of worms. a** Autophagosomes (APs, yellow arrowheads) and autolysosomes (ALs, white arrowheads) in the intestine cells (dashed lines) of indicated strains at day 1 of adulthood post 1 h of 5 mM chloroquine (CQ) or mock treatment. $n =$ 3 independent experiments with at least 20 worms. Scale bar: 10 μm. $n$ indicates number of animals. Exact sample size and $p$ value are included in Source Data file. **b** The thrashing rates (a metric for motility) of the indicated worms at day 1 and 10 of adulthood. $n$ indicates number of animals. Exact sample size and $p$ value are included in Source Data file. **c** Overexpressing *daf-19c* reduces polyQ-YFP aggregates (arrowheads) in body wall muscle. Scale Bar: 100 μm. $n = 3$ biological independent experiments. Exact sample size and $p$ value are included in Source Data file. **d** Worms with faster IFT at day 10 of adulthood live longer. Worms were ranked by the velocities of the indicated IFT components from the fastest to the slowest. $n$ indicates number of animals. Exact sample size and $p$ value are included in Source Data file. **e** Overexpressing *daf-19c* extends lifespan. $n = 4$ biological independent experiments. Exact sample size and $p$ value are included in Source Data file. **f, g** Survival curves of WT worms and the worms overexpressing *daf-19c* undergoing indicated RNAi treatments. Note that disrupting cilia (**f**) or blocking AMPK signalling (**g**) abolishes the *daf-19c*-induced longevity. $n = 3$ biological independent experiments. Exact sample size and $p$ value are included in Source Data file. **h**. A graphic summary. In aged worms, the reduced IFT in sensory cilia blunts sensory perception and dysregulates AMPK signalling first in sensory neurons through *par-4*/LKB1 and in turn in other tissues such as the intestine. Overexpressing *daf-19c* enhances IFT, improves sensory perception and in turn activates AMPK and autophagy. See discussion for details. Data are presented as mean ± SD. Poisson regression in **a**, one-way ANOVA in **b**, **c**, unpaired $t$-test (two-tailed) in **d**, $*p < 0.05$, $**p < 0.01$, $***p < 0.001$, ns non-significant. See Source Data for the lifespan statistics in **e**–**g**. Source data are provided as a Source Data file.

gift from Ou Lab. To generate *L3781-Prab-3::daf-19c::mCherry*, *daf-19c* promoter was replaced by 1357 bp of *rab-3* promoter in *Pdaf-19c::daf-19c::mCherry*. To generate *L3781-Pdyf-1::sid-1*, 454 bp of *dyf-1* promoter amplified from N2 genomic DNA and 2382 bp of *sid-1* cDNA amplified from N2 cDNA were cloned into *L3781* using Gibson Assembly[48].

For fluorescent tag knock-in, plasmids were constructed as described[49]. Briefly, sgRNAs for the target genes were selected from Zhang lab's CRISPR design tool at http://crispr.mit.edu and inserted into pDD162 (a gift from Bob Goldstein, Addgene #47549). Homology recombination templates were constructed by cloning the ~0.6 kb of 5′ and 3′ homology arms into pDD282 and pDD284 plasmids (gifts from Bob Goldstein, Addgene # 66823) using NEB Gibson Assembly kit. Target sites in the templates were modified with synonymous mutations. All tags were inserted at the C-terminal of genes of interest.

**Transgenes**. Extra-chromosomal transgenic lines of *daf-19c::mCherry* were obtained by co-injecting the plasmids of *L3781-Pdaf-19c::daf-19c::mCherry* and *Pmyo-3::CFP* or *Pegl-17::mCherry*, *L3781-Prab-3::daf-19c::mCherry* and *Pegl-17:: mCherry* into N2. Plasmid concentrations for microinjections were 50 ng/μl for the genes of interest and 20 ng/μl for injection marker, respectively.

For knock-in lines, injections and subsequent screens were performed as described[49]. Self-excising selection cassettes were discarded before sequencing and phenotypic analysis. The *par-4(syb1018)V* allele was generated by SunyBiotech using CRISPR/Cas9 technology. mNeonGreen-3xFLAG were inserted into the C-terminal of the endogenous *par-4* gene. All knock-in strains were verified by DNA sequencing.

**RNA interference**. RNAi experiments were performed using *E.coli* HT115 bacteria on standard NGM plates containing 100 μg/ml ampicillin and 0.8 mM IPTG as described[50]. Worms were grown on HT115 expressing dsRNA against indicated genes from the egg until the corresponding time unless otherwise noted. HT115 expressing dsRNA against *luc2*, a real but not worm gene, served as a control to minimise off-target phenotypes. The strain of HT115 [L4440::luc2] was a gift from Antebi lab (MPI-AGE). For neuron-specific RNAi, TU3401 worms were used directly or after crossed with indicated strains. For sensory neuron-specific RNAi, SYD0779 (*sid-1(pk3321) V; sydEx196[pdyf-1::sid-1, egl-17p::mCherry]*) were used directly or after crossed with indicated strains.

**Quantitative RT-PCR**. More than 100 well-fed synchronised worms were collected into QIAzol reagent (QIAGEN), and column purified by RNeasy Mini (QIAGEN). cDNA was subsequently generated by iScriptTM Reverse Transcription Supermix for RT-qPCR (Bio-Rad). Quantitative RT-PCR was performed with Bestar® Sybr Green qPCR Master Mix (DBI Bioscience) or 2xNovoStart® SYBR qPCR SuperMix Plus (Novoprotein) on a QuantStudioTM 6 Flex Real-time PCR System (Applied Biosystems) or a CFX384 TouchTM Real-Time PCR Detection System (Bio-Rad). mRNA levels of *ama-1* and *cdc-42* were used for normalisation. Four technical replicates were performed in each reaction. At least three biological repeats were examined. Primer sequences are listed in Supplementary Data 2.

**Microscopy**. Live imaging of intraflagellar transport was performed following a previous report[51]. In brief, on the same day, worms at different ages were anaesthetised with 5 mM levamisole in M9 buffer, mounted on 5% agar pads and maintained at room temperature. Images were collected using an Olympus IX81 microscope equipped with a ×100, 1.49 NA objective and an Ultraview spinning disc confocal head (PerkinElmer Ultra VIEW VoX). Time-lapse images were acquired at an exposure time of 200 ms for 30 s (spinning disk). Cilia were chosen based on their orientation plane, with the base, proximal segment and distal segment in focus.

Fluorescence images were obtained using a Leica TCS SP8 confocal microscope or an Olympus BX53 microscope. Animals were anaesthetised using 5 mM levamisole and mounted on 5% agar pads. Fluorescent intensities were measured by ImageJ[52]. For fluorescence quantification of worms at different ages, worms at day 1 and day 10 of adulthood were examined on the same day using the same microscope. In each independent assay, every young and old worm's fluorescent intensity was further normalised against the mean of that in day-1 old worms.

For the analysis of the subcellular localisation of CRTC-1::RFP, the mean nuclear and cytoplasmic fluorescent intensities of RFP were measured and subjected to the calculation for the nucleo-cytoplasmic ratio.

For the analysis of abnormalities in body wall muscle, MYO-3::GFP in the body wall muscle was imaged. The abnormalities were characterised into two types: a general disorganisation of the myofilaments with GFP aggregations and gaps in the lattice[33].

For the analysis of polyQ strains, the numbers of polyQ aggregates in the body wall muscle were counted in individual worms at day 3 of adulthood. For each genotype, at least 56 animals from three independent experiments were scored.

**Kymograph generation and analysis**. Kymographs were generated and analysed as described[51]. In brief, Fourier filtered and separated anterograde or retrograde kymographs were generated with the KymographClear toolset plugin in ImageJ (http://www.nat.vu.nl/~erwinp/downloads.html). IFT velocities at every 0.5 μm along cilia were measured by the KymographDirect software (http://www.nat.vu.nl/~erwinp/downloades.html) from the kymographs. Velocity curves were subsequently generated using GraphPad Prism (GraphPad Software). Cilia with projection lengths smaller than 7 μm were not examined.

**Autophagy analysis**. mCherry::GFP::LGG-1 puncta in posterior intestinal cells were counted from about 5 slices with 1 μm step size, the Z-position was selected where intestinal nucleus could be seen clearly. The puncta were counted using ComDet v.0.3.7 in ImageJ. For each genotype, at least 20 worms at corresponding stages from three independent experiments were scored. The number of APs was calculated by the GFP-positive puncta, and the number of ALs was calculated by the puncta with only mCherry signal.

**Western blotting**. Synchronised worms were grown to indicated ages and collected in M9. To test the effect of food perception on *daf-19c*-induced p-AMPK, worms at day 1 of adulthood were washed off plates and further washed three times by M9 buffer before transferred to prepared 90-mm plates for the test. Three groups of plates were set up: (a) plates seeded with OP50 (food and odour); (b) empty plates with OP50 only on lids (odour); (c) empty plates (no food or odour). After 1 or 12 h, worms were collected in M9 for western blotting to test p-AMPK level.

After three rounds of washing with M9, 4x SDS gel-loading buffer (Takara, Cat#9173) was added into worm samples and kept at −80 °C. Proteins were separated by reducing SDS-PAGE and transferred to PVDF membranes. Membranes were then blotted with antibodies against p-AMPK (CST, Cat# 4188 s, dilution: 1:1,000), α-tubulin (Sigma-Aldrich, Cat# T5168, dilution: 1:2,000). An anti-mouse secondary antibody conjugated with horseradish peroxidase (Life Technologies, Cat# G21040, 1:5,000) was used for detecting anti-α-tubulin, and an anti-rabbit secondary antibody (Life Technologies, Cat# G21234, 1:5000) was used for detecting anti-p-AMPK primary antibodies. Signals of western blotting were measured using Adobe Photoshop. Background signals were subtracted as reported[53].

**Dye-filling assay**. DiI staining was performed as described with modifications[54]. Approximately 20–30 day 1 or day 10 adult worms were randomly picked into 200-

µl M9 solution. After washing with M9 for three times to remove bacterial, worms were incubated with 1 µg/ml fluorescent Dye (DiI 1,1′-dioctadecyl-3,3,30,30,-tetramethylindo-carbocyanine perchlorate, Sigma) in dark at room temperature for 30 min. Worms were subsequently washed with M9 and transferred to regular NGM plates for 30 min. For imaging by an Olympus BX53 microscope, worms were mounted on 5% agarose pads and anaesthetised with 5 mM levamisole. At least three independent assays were performed.

**Auxin treatment**. Auxin treatment was performed by transferring worms to bacteria-seeded plates containing auxin as reported[26]. The natural auxin indole-3-acetic acid (IAA) (Alfa Aesar, #A10556) was prepared as a 400-mM stock solution in ethanol. Auxin was diluted into the NGM agar at 1 mM.

**Enhanced slowing response assay**. Enhanced slowing responses (ESR) were assayed as reported with modification[55]. About 20 food-deprived worms were washed free of bacterial in M9 buffer and transferred to NGM plates with no bacteria or a ring-like bacterial lawn in the middle. They were incubated on these plates for 30 min at room temperature before their body bends were recorded. For the worms on the plates with bacteria, only those on the bacteria lawn were scored.

**Chemotaxis assay**. Chemotaxis assay was performed as described with modification[56]. In brief, on a 6-cm unseeded plate, 1 µl of 1 M $NaN_3$ were spotted at the odourant spot with 1 µl of 10% butanone or nonanone and the control spot with 1 µl of 95% ethanol freshly before assay. Around 200 synchronised worms at indicated ages were placed at the centre of the plate (origin) and recorded by a Leica camera for their movement for 1 h. Chemotaxis index (CI) was calculated at the end of the assay as: $CI = (N_{butanone/nonanone} - N_{ethanol})/(N_{total} - N_{origin})$. The worms were counted by the ComDet plugin in ImageJ (https://github.com/ekatrukha/ComDet/wiki).

**Motility assay**. Worms were transferred into a M9-filled 96-well plate with one in each well and recorded using an Olympus SZX16 stereomicroscope equipped with a Nikon D4 camera. The thrashing rate was subsequently scored from videos. For each genotype or treatment, around 20 worms were examined in each of the three replicates.

**Statistical analysis**. Results are presented as Mean ± SD unless otherwise noted. Statistical tests were performed as indicated using GraphPad Prism (GraphPad Software). Detailed statistical information is shown in Source data.

**Reporting summary**. Further information on research design is available in the Nature Research Reporting Summary linked to this article.

## Data availability
All data are available within the Article and Supplementary Files, or available from the corresponding author upon reasonable request. A reporting summary for this Article is available as a Supplementary Information file. Source data are provided with this paper.

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

## Acknowledgements

We thank Dr. Jindong Han (Peking U.) for plasmids, Dr. Adam Antebi (MPI-AGE) and CGC for strains, Drs. Guangshuo Ou (Tsinghua U.) and Yixian Zheng (Carnegie Institution for Science) for helpful discussion and Mr. Chenghui Wan for illustration.

This research was supported by National Natural Science Foundation of China (91749119, 31900503, 31991193), the Strategic Priority Research Program of the Chinese Academy of Sciences (XDB19000000) and the Thousand Talents Plan (Youth).

## Author contributions

Y.Z., X.Z., X.Y. and Y.S. conceived the project and designed the experiments. Y.Z. and X.Z. performed experiments and analysed data, with the assistance of Y.D. and M.S.. Y.Z. performed autophagy assays. J.Z. helped in live imaging microscopy. Y.Z., X.Y. and Y.S. wrote the manuscript. All authors contributed to manuscript editing.

## Competing interests

The authors declare no competing interests.
