## [Peer Review File · Nature Communications]

Reviewers' Comments:

Reviewer #1:

Remarks to the Author:

Zhang et al. described IFT decrease impairs sensory perception and metabolism in ageing of *C. elegans*. Zhang et al. found that IFT decreased in aged worms, which is dependent on the TF DAF-19 activity. The authors showed that the decrease of sensory perception in aged worms suppressed AMPK signaling across tissues, which is a novel and interesting finding. They further overexpressed DAF-19 and promote fitness and longevity. Overall, the study was well performed and the writing is clear. I support the publication of this work in Nature Communication when the following minor issues can be resolved. Most of them do not need additional experiments.

1. Figure 1C, the ciliary middle and distal segment are not in the correct ratio.
2. I would not conclude that IFT is impaired during ageing. Impairment is an overstatement. Reduction might be a more appropriate description. In addition, authors should point out that both anterograde and retrograde IFT are slowed down. why the authors measure bidirectional IFT using different markers?
3. Figure 1F, it seems that CHE-11 moves faster than CHE-3. How to interpret that cargo moves faster than its motor? The same question is for figure 3C.
4. Figure 2 relied on fluorescence quantifications extensively at different dates. A fiducial marker is essential.
5. Figure 3C, why OSM-3 was not enhanced?
6. Ciliated sensory neurons are notorious for their resistance to RNAi, despite the use of RNAi sensitive background. Some of the key results relied on RNAi in ciliated neurons. Did the author checked the loss of protein or Dyf phenotype in their RNAi experiments?

Reviewer #2:

Remarks to the Author:

Primary cilia are sensory/signaling organelles found on many cell types in metazoans, including neurons, and are implicated in a variety of cell and organismal functions. They consist of a microtubule-based axoneme whose synthesis depends on a master RFX transcriptional regulator and an intraflagellar transport (IFT) system. Their importance is evident when considering its association with numerous human disorders.

This study by Zhang and colleagues uncovers the role of aging and the master ciliary regulator on the IFT system, sensory perception, metabolic functions, and autophagy. Aside from a few minor issues, listed below, the studies are very well executed and of great interest. The scope of the studies is impressive. For example, to assess the role of the RFX factor DAF-19 in homeostasis and health, the authors examine the accumulation of polyglutamine aggregates, something that is typically done in studies that specifically address protein homeostasis and neurodegeneration, for example. The authors also address the status of IFT during aging, and show a correlation between IFT and lifespan. Finally, they show a role for sensory perception and AMPK signaling in motility, another indicator of healthy aging.

In brief, this study represents one of the most detailed looks at the relationship between aging, the IFT system required to build and maintain proper ciliary function, and other determinants of health (sensory perception, motility, etc.) It is very well written, the data are of high quality, and will be of interest to a wide audience. I encourage its publication in Nature Communications.

MINOR ISSUES:

109. The authors use an assay where the movement of worms is monitored on food, termed an

enhanced slowing response (ESR) to bacteria (Fig. 1a and Supplementary Fig. 1a). They claim that older worms show a difference compared to young worms, indicating a defect of sensory perception with ageing. This is fine, but this is not necessarily a standard way to analyse chemosensory responses in animals with presumed cilia defects. The authors should conduct standard chemotaxis assays (showing attraction and repulsion to specific compounds) to complement and provide more weight to their results. One chemotaxis assay was performed in Figure 3, to compare WT and *daf-19c* oe strains. Chemotaxis assays should be presented here during different developmental stages.

113. DiI staining differences may not be indicative of cilia structure defects; anomalies in sheath cells can also cause a dye-filling defect. A dye-filling defect is therefore not consistent with the authors' statement that "staining of DiI in the soma of sensory neurons became remarkably weaker at D10 (Fig. 1b), showing that ageing causes defects in the sensory cilia.". The authors should investigate the structure of cilia more directly before they can claim a cilium structure defect. One way would be to use a non-IFT ciliary reporter (e.g., diffuse GFP which can enter the cilium) to measure the length of amphid and phasmid cilia as the animals age. Shorter cilia could be the cause of the dye-filling defect. Notably, the authors mentioned on line 116 that it has been shown that the "ciliary structure is unaffected by ageing at D10 (day 10)", directly contradicting their previous statement.

125. If aged worms have slightly longer cilia (as measured by IFT reporters) this is not necessarily consistent with a dye-filling defect.

171. Figure 3e. Why are there connections between WT and *daf-19c* oe strains? It's unclear what comparison is made here. Also, there are only two data points for WT, which is insufficient, and three for the oe strain, which is just sufficient.

243. Throughout the manuscript, the authors should make an effort to state which model system(s) certain statements apply to. For example, "Activated AMPK...." doesn't specify: is this true in worms? The title of the cited paper doesn't necessarily specify (as is the case here)

265. myofilament abnormalities (remove 's' from myofilament)

Reviewer #3:

Remarks to the Author:

The manuscript entitled "The decrease of intraflagellar transport impairs sensory perception and metabolism in ageing" reports the role of DAF-19/RFX in aging-associated decline in sensory functions and in aging and health. The authors showed that intraflagellar transport (IFT) of sensory cilia displayed an age-dependent impairment. They found that the level of DAF-19/RFX transcription factor, key regulator of IFT gene expression, decreased in neurons with age. They showed that overexpressing *daf-19* enhanced the IFT in aged worms. The *daf-19* overexpression increased AMPK activity in neurons and the intestine via PAR-4/LKB1 kinase and octopamine signaling. They found that the activation of AMPK by *daf-19* overexpression upregulated autophagy and extended lifespan. The findings are interesting and are very useful for the field of aging research. However, I find many concerns regarding this paper.

Major comments

1) The biggest issue I found in this paper is they mainly used *daf-19* overexpression for their claims. If available *daf-19* mutants have cilia development complications, they can use degron knockin with adult only auxin treatment AND neuron-specific *daf-19* RNAi to make sure ALL their main findings with *daf-19* overexpression are conversely changed with the degron knockin and RNAi.

2) How does *daf-19* overexpression upregulates PAR-4/LKB1? That part will be the most important

mechanistic point and missing.

3) The *daf-19*-overexpressing animals shown in this paper are extra-chromosomal transgenic lines. I think this is an issue too because all their findings rely on the transgenic animals. On page 8, they say "Mild overexpression of *daf-19c*,..." and this is not scientific and what is the basis of this? In addition, the authors used WT(N2) for control experiments. The authors need to confirm whether the co-injection markers they used (*myo-3p::cfp*, *egl-17p::mCherry*, or *egl-17p::rfp*) had no impact on their key physiology experiments. Overall, the complete dependency of their main findings on the extra-chromosomal transgenic lines weakens the findings in this paper.

4) On page 16, the authors say, "Food perception without ingestion is known to drive insulin/IGF-1 signalling (IIS). Our results indicate that it also induces AMPK signalling and in turn autophagy. Therefore, food perception contributes to metabolic homeostasis before food intake by regulating the pivotal regulators of both anabolism (IIS) and catabolism (AMPK) (Fig. 6h)." This part is confusing to readers who know the relationship between IIS and AMPK, and their roles in autophagy and lifespan, because what these sentences say are actually conflicting with each other. There are many places like this in this paper. This is mainly because their major findings with *daf-19* overexpression to AMPK activation to autophagy to longevity is conflicting with many previous reports showing sensory defects to reduction of insulin signaling (to perhaps AMPK activation and autophagy) to health/motility and longevity. I think the authors tried to explain this issue in the paper but unfortunately it is still very confusing as it is. I think they need to rewrite the paper substantially to resolve this confusion, and that will be very important for improving this paper.

5) The authors demonstrated that enhancing sensory cilia functions via overexpressing *daf-19* upregulated AMPK activity by showing increased p-AMPK level. Considering that *daf-19* was downregulated during aging, and sensory neural functions were also deteriorated, as the authors depicted in the Fig. 6h, one can expect that p-AMPK level is decreased during aging. However, Fig. 4b, 4e, and S4c all showed that p-AMPK level increased in old worms compared to young worms, and no description about this discrepancy is written in the manuscript. The authors should describe the age-dependent change in p-AMPK level. Otherwise it does not explain the authors' suggested model.

6) In Fig. 2d, they need to show the data and the quantification for proteins in cilia, as in Fig. 2a to support their claim.

7) The authors should provide the raw data of all Western blot they present in this study in supplementary materials with proper size markers labeled.

Minor concerns

1) The authors used day 10 worms of lifespan assays for IFT velocity and used them again for lifespan assay after recovery in Fig. 6d. Does the exposure to levamisole at day 10 affect the result of lifespan assay?

2) It would be better if authors guide readers to look into supplementary table S3 for the number of independently repeated experiments, total number of worms counted, and the statistical analysis of the presented data in the figure legends.

3) On page 8, the authors described that "a mild *daf-19* RNAi did not disrupt ciliogenesis but abolished the elevated motilities of OSM-3::GFP and OSM-6::GFP in *daf-2(-)* mutants at D10 to WT levels and substantially downregulated the velocity of CHE-3::GFP and CHE-11::GFP in *daf-2(-)* mutants (Supplementary Fig. 2d-2f)." However, in Fig. S2e, *daf-19* RNAi did not reduce the motility of OSM-3::GFP or OSM-6::GFP in *daf-2(-)* mutants at day 10, as there are no significant changes between *daf-2(-)* *luc2* RNAi and *daf-2(-)* *daf-19* RNAi animals. The authors should revise the manuscript to provide proper explanation of the data.

4) The authors should provide detailed explanation for the criteria of quantifying CRTC-1::RFP subcellular localization shown in Fig. 4c.

5) What does the percentage mean in the Fig. 6d? The authors should provide detailed explanation of the criteria for dividing fast and slow IFT-displaying worm groups.

6) There is no description for Fig. S2f in the figure legends.

7) On page 9, The authors described, "IFT mutants of *osm-3(-)* and *osm-6(-)* exhibited severe cilia defects and IIS target genes were regulated as reported at D1 (Supplementary Fig. 3a). In aged worms, IIS exhibited a decreased modulation by sensory perception, as multiple IIS target genes

were no longer changed upon mutating *osm-3* or *osm-6* at D10 (Supplementary Fig. 3a).” I think the authors made these sentences very vague (perhaps intentionally) because *osm-3(-)* and *osm-6(-)* actually UPregulate DAF-16 target genes, which contribute to longevity as many previous papers reported. However, I think this will make many readers even more confused, and I suggest that the authors revise the sentences to incorporate the direction of gene expression changes. If necessary, the authors may discuss this confusing issue in the discussion further.

8) Have the authors used neuron-specific RNAi strain backgrounds (TU3410 or SYD0779) for lifespan assays? The authors should then provide detailed strain information in the supplementary table 3, lifespan analysis tab to provide precise information regarding strains they used.

Our detailed responses to the reviewers' concerns are listed below on a point-by-point basis.

Reviewer #1 (Remarks to the Author):

Zhang et al. described IFT decrease impairs sensory perception and metabolism in ageing of *C. elegans*. Zhang et al. found that IFT decreased in aged worms, which is dependent on the TF DAF-19 activity. The authors showed that the decrease of sensory perception in aged worms suppressed AMPK signaling across tissues, which is a novel and interesting finding. They further overexpressed DAF-19 and promote fitness and longevity. Overall, the study was well performed and the writing is clear. I support the publication of this work in Nature Communication when the following minor issues can be resolved. Most of them do not need additional experiments.

1. Figure 1C, the ciliary middle and distal segment are not in the correct ratio.

-- Thank you for your comment! The figure has been modified accordingly.

2. I would not conclude that IFT is impaired during ageing. Impairment is an overstatement. Reduction might be a more appropriate description. In addition, authors should point out that both anterograde and retrograde IFT are slowed down.

-- The manuscript has been modified following your advice.

why the authors measure bidirectional IFT using different markers?

-- This is because we aimed to show IFT by the proteins regulating anterograde and

retrograde transportation, respectively. In the revised manuscript, the bidirectional motilities of each examined IFT component were shown following your comment (Figure 1d-f, Figure 3a-c, Supplementary Figure 2a-c, Supplementary Figure 3c-e).

3. Figure 1F, it seems that CHE-11 moves faster than CHE-3. How to interpret that cargo moves faster than its motor? The same question is for figure 3C.

-- Although the same IFT components were examined on a same day, as our reply to your 4th major concerns, different molecules were live imaged on different dates. Therefore, this difference could be due to the fluctuations in experiment conditions. The same could happen to OSM-6 and OSM-3 in Figure 3. Because the focus of this study is on the age-related changes on IFT but not on the different motilities among IFT components, we think this does not affect the scientific conclusions in this manuscript.

4. Figure 2 relied on fluorescence quantifications extensively at different dates. A fiducial marker is essential.

-- Thanks for your comment! In each biological replicate of fluorescence quantification in Figure 2, day-1 and day-10 old worms were examined on the same microscope and on the same day (not at different dates). The fluorescent intensity was further normalised against the mean of that in day-1 old worm in each replicate. Therefore, a fiducial marker is not necessary. We have clarified this issue in the revised manuscript.

5. Figure 3C, why OSM-3 was not enhanced?

-- In Figure 3, the anterograde motility (i.e., velocity and frequency) of OSM-3 is not

enhanced in young worms, as you noted, but exhibits a clear increase in aged worms upon *daf-19c* upregulation. The other IFT components were increased by *daf-19c* upregulation in both young and old worms. Unlike in aged worms, IFT in young worms are well maintained. That could be the reason why upregulating *daf-19c* has a bigger impact in aged worms. Besides, these data imply a complex regulation of *daf-19c* on different IFT components, potentially because *daf-19c* drives IFT genes expression at different levels. This will be intriguing to explore in the future. Corresponding discussion has been included in the revised manuscript.

6. Ciliated sensory neurons are notorious for their resistance to RNAi, despite the use of RNAi sensitive background. Some of the key results relied on RNAi in ciliated neurons. Did the author check the loss of protein or Dyf phenotype in their RNAi experiments?

-- Many thanks for your critics! We checked the RNAi efficiency in sensory neurons following your suggestion. GFP signal in sensory neurons from *osm-6p::gfp* is significantly suppressed upon *gfp* RNAi in the *sid-1;uls69* background (Supplemental Figure 3a). Consistently, the transcription of IFT genes, which are *daf-19* targets and specifically expressed in ciliated sensory neurons, are suppressed upon *daf-19* RNAi (Supplemental Figure 3b). These data therefore confirmed the effectiveness of RNAi in our assays. Besides, RNAi is reported to work in these neurons in *sid-1;uls69*¹.

Reviewer #2 (Remarks to the Author):

Primary cilia are sensory/signaling organelles found on many cell types in metazoans, including neurons, and are implicated in a variety of cell and organismal functions. They consist of a microtubule-based axoneme whose synthesis depends on a master

RFX transcriptional regulator and an intraflagellar transport (IFT) system. Their important is evident when considering its association with numerous human disorders.

This study by Zhang and colleagues uncovers the role of aging and the master ciliary regulator on the IFT system, sensory perception, metabolic functions, and autophagy. Aside from a few minor issues, listed below, the studies are very well executed and of great interest. The scope of the studies is impressive. For example, to assess the role of the RFX factor DAF-19 in homeostasis and health, the authors examine the accumulation of polyglutamine aggregates, something that is typically done in studies that specifically address protein homeostasis and neurodegeneration, for example. The authors also address the status of IFT during aging, and show a correlation between IFT and lifespan. Finally, they show a role for sensory perception and AMPK signaling in motility, another indicator of healthy aging.

In brief, this study represents one of the most detailed looks at the relationship between aging, the IFT system required to build and maintain proper ciliary function, and other determinants of health (sensory perception, motility, etc.) It is very well written, the data are of high quality, and will be of interest to a wide audience. I encourage its publication in Nature Communications.

MINOR ISSUES:

109. The authors use an assay where the movement of worms is monitored on food, termed an enhanced slowing response (ESR) to bacteria (Fig. 1a and Supplementary Fig. 1a). They claim that older worms show a difference compared to young worms, indicating a defect of sensory perception with ageing. This is fine, but this is not necessarily a standard way to analyse chemosensory responses in animals with presumed cilia defects. The authors should conduct standard chemotaxis assays (showing attraction and repulsion to specific compounds) to complement and provide more weight to their results. One chemotaxis assay was performed in Figure

3, to compare WT and *daf-19c* oe strains. Chemotaxis assays should be presented here during different developmental stages.

-- Thanks for your suggestion. Worm larvae are not suitable for chemotaxis assay since their neural system is different from adults². Therefore, we performed chemotaxis assays of both attraction and repulsion in day-1 and day-5 old worms (Figure 3e). These assays show that upregulating *daf-19c* inhibits the decrease of sensory perception in aged worms.

113. Dil staining differences may not be indicative of cilia structure defects; anomalies in sheath cells can also cause a dye-filling defect. A dye-filling defect is therefore not consistent with the authors' statement that "staining of Dil in the soma of sensory neurons became remarkably weaker at D10 (Fig. 1b), showing that ageing causes defects in the sensory cilia."

-- Thanks for your critics! We agree with your comment. The corresponding statement is modified as "staining of Dil in the soma of sensory neurons became remarkably weaker at D10 (Fig. 1b), suggesting that ageing causes..." in the revised manuscript.

The authors should investigate the structure of cilia more directly before they can claim a cilium structure defect. One way would be to use a non-IFT ciliary reporter (e.g., diffuse GFP which can enter the cilium) to measure the length of amphid and phasmid cilia as the animals age. Shorter cilia could be the cause of the dye-filling defect.

-- We measured cilia length with a diffusive GFP reporter (*osm-6p::gfp*) as you suggested (Supplemental Figure 1b) and found that cilia length does not change in worms at day 10 of adulthood. Therefore, the slight changes of CHE-3::GFP and CHE-11::GFP signal (Supplemental Figure 1c-1d) could be due to dysregulation of IFT

in aged worms.

Notably, the authors mentioned on line 116 that it has been shown that the “ciliary structure is unaffected by ageing at D10 (day 10)”, directly contradicting their previous statement.

-- Sorry for any misunderstanding. Defects in the sensory cilia could be from cilia structure, IFT, or other regulators. In this study, we found that the sensory cilia defects in day-10 old worms is not from changes in structure, but from decreased IFT. Therefore, this is not contradicting our previous statement. We have further clarified our statement in the revised manuscript.

125. If aged worms have slightly longer cilia (as measured by IFT reporters) this is not necessarily consistent with a dye-filling defect.

-- We agree with your comment. But we think this could be a clue suggesting disrupted IFT, which could cause dye-filling defect, since there are reports that IFT could affect cilia^{3,4}.

171. Figure 3e. Why are there connections between WT and *daf-19c* oe strains? It's unclear what comparison is made here. Also, there are only two data points for WT, which is insufficient, and three for the oe strain, which is just sufficient.

-- Sorry for the confusion. The connections are to show the pairs of WT and *daf-19c* oe data points in three independent assays. There are three data points for both WT and *daf-19c* oe strain. However, two of the WT points are too close to distinguish in the figure (0.155779 in replicate 2 and 0.159341 in replicate 3, Figure 1 for reviewer).

Figure 1 for reviewer. Chemotaxis index of WT and *daf-19c* oe worms at day 5 of adulthood.

We listed the corresponding data in Supplementary Table 3. In the revised manuscript, to avoid misunderstanding and to improve readability, we presented the chemotaxis data in a different form of graph (Figure 3e). Moreover, following your suggestion to examine worms' chemotaxis at different ages (109), we re-performed the assay so that worms at D1 and D5 were examined together.

243. Throughout the manuscript, the authors should make an effort to state which model system(s) certain statements apply to. For example, "Activated AMPK..." doesn't specify: is this true in worms? The title of the cited paper doesn't necessarily specify (as is the case here)

-- Many thanks for your critics! We have modified the manuscript following your comment. AMPK is widely accepted as a positive regulator of longevity in *C. elegans* as reported in "Apfeld, J., O' Connor, G., McDonagh, T., DiStefano, P. S. & Curtis, R. The AMP-activated protein kinase AAK-2 links energy levels and insulin-like signals to lifespan in *C. elegans*. *Genes Dev.* 18, 3004 – 3009 (2004)". We have changed the cited paper accordingly.

265. myofilament abnormalities (remove 's' from myofilament)

-- We appreciate your careful reading. The mistake has been corrected in the revised manuscript.

Reviewer #3 (Remarks to the Author):

The manuscript entitled "The decrease of intraflagellar transport impairs sensory perception and metabolism in ageing" reports the role of DAF-19/RFX in

aging-associated decline in sensory functions and in aging and health. The authors showed that intraflagellar transport (IFT) of sensory cilia displayed an age-dependent impairment. They found that the level of DAF-19/RFX transcription factor, key regulator of IFT gene expression, decreased in neurons with age. They showed that overexpressing *daf-19* enhanced the IFT in aged worms. The *daf-19* overexpression increased AMPK activity in neurons and the intestine via PAR-4/LKB1 kinase and octopamine signaling. They found that the activation of AMPK by *daf-19* overexpression upregulated autophagy and extended lifespan. The findings are interesting and are very useful for the field of aging research. However, I find many concerns regarding this paper.

Major comments

1) The biggest issue I found in this paper is they mainly used *daf-19* overexpression for their claims. If available *daf-19* mutants have cilia development complications, they can use degron knockin with adult only auxin treatment AND neuron-specific *daf-19* RNAi to make sure ALL their main findings with *daf-19* overexpression are conversely changed with the degron knockin and RNAi.

-- Many thanks for your critics and suggestions! Following your suggestions, we have performed neuron-specific RNAi against *daf-19* in adult worms. Consistent with our main findings with *daf-19c* overexpression, suppressing *daf-19* in neurons downregulates p-AMPK (Figure 4c) and impairs the integrity of myofilaments in aged worms (Supplementary Figure 7d). Another main finding in IFT had already been confirmed by neuron-specific RNAi against *daf-19* in the previous version of our manuscript (Supplementary Figure 3). Inhibiting *daf-19* in neurons decreases the motilities of multiple IFT components.

As for the degron-auxin system, degron knock-in strains with TIR1 expressed specifically in neurons have to be made to achieve similar inhibitory effect as

neuron-specific RNAi against *daf-19*. This is indeed time consuming. Therefore, in the limited time for revision, we struggled to finish *daf-19* loss-of-function assays using the RNAi method. We believe that the results from RNAi assays are enough to show that inhibiting *daf-19* conversely affects the phenotypes caused by overexpressing *daf-19c*.

2) How does *daf-19* overexpression upregulate PAR-4/LKB1? That part will be the most important mechanistic point and missing.

-- Good question! In this study, we found that upregulating *daf-19* increases IFT. Previous studies indicated that the mammal ortholog of PAR-4, LKB1, is activated in cilia and in turn switches on AMPK signalling⁵. Therefore, IFT, the machinery transporting molecules in and out of cilia, must be involved in the ciliary localization of LKB1. Consistent with this hypothesis, we show in the revised manuscript that overexpression *daf-19c* upregulates the ciliary localization of PAR-4 (Figure 5b). Moreover, several cilia proteins have been found to control the ciliary localization of LKB1^{6,7}. Among them, we found that *flcn-1*, the ortholog of mammal FLCN (folliculin), regulates the *daf-19c*-induced ciliary localization of PAR-4 and activation of AMPK (Supplementary Figure 6c). These results suggest that *daf-19c* controls PAR-4 through a “*daf-19c* – IFT – *flcn-1* – *par-4*” axis.

3) The *daf-19*-overexpressing animals shown in this paper are extra-chromosomal transgenic lines. I think this is an issue too because all their findings rely on the transgenic animals. On page 8, they say “Mild overexpression of *daf-19c*,..” and this is not scientific and what is the basis of this?

-- Sorry for the confusion! By qPCR, we found that the transgene overexpresses *daf-19c* by ~3-fold of the endogenous level (Supplementary Figure 4b). To minimize biased results from a certain extra-chromosomal transgene, we also used three independent transgenes (two with its native promoter and one with a

neuron-specific promoter).

In addition, the authors used WT(N2) for control experiments. The authors need to confirm whether the co-injection markers they used (*myo-3p::cfp*, *egl-17p::mCherry*, or *egl-17p::rfp*) had no impact on their key physiology experiments.

-- We examined the potential impact of these injection markers as suggested. They do not affect p-AMPK levels (Supplementary Figure 5a) or worms' motility (Supplementary Figure 7a).

Overall, the complete dependency of their main findings on the extra-chromosomal transgenic lines weakens the findings in this paper.

-- We agree with your critics. Following your suggestion to suppress *daf-19*, we have shown the function of *daf-19* from two perspectives in the revised manuscript.

4) On page 16, the authors say, "Food perception without ingestion is known to drive insulin/IGF-1 signalling (IIS). Our results indicate that it also induces AMPK signalling and in turn autophagy. Therefore, food perception contributes to metabolic homeostasis before food intake by regulating the pivotal regulators of both anabolism (IIS) and catabolism (AMPK) (Fig. 6h)." This part is confusing to readers who know the relationship between IIS and AMPK, and their roles in autophagy and lifespan, because what these sentences say are actually conflicting with each other. There are many places like this in this paper. This is mainly because their major findings with *daf-19* overexpression to AMPK activation to autophagy to longevity is conflicting with many previous reports showing sensory defects to reduction of insulin signaling (to perhaps AMPK activation and autophagy) to health/motility and longevity. I think the authors tried to explain this issue in the paper but unfortunately it is still very confusing as it is. I think they need to rewrite the paper substantially to resolve this confusion, and that will be very important for improving this paper.

-- We do appreciate that you notice this “conflict” and our effort to explain it. We have further explained this issue in the revised “discussion” section.

In brief, this “conflict” reflects the two facets of metabolism, the anabolism driven by insulin/IGF signalling (IIS) and the catabolism driven by AMPK. Previous studies discover the role of sensory perception in the former facet, whereas this manuscript shows a direct regulation of sensory perception on the latter one. These two facets “conflicts” with each other to maintain the balance of metabolism. Food perception simultaneously controls both facets for metabolic homeostasis. Steering cars with two hands is always safer than with one hand.

As sensory perception controls two facets of metabolism, either disrupting or improving cilia function extends lifespan, as previously reported and discovered by our study. IIS has a wide range of downstream effectors other than AMPK^{8,9}. The IIS defective *daf-2(-)* mutants with reduced AMPK activity are still longer-lived than WT worms¹⁰. Therefore, sensory mutants with reduced AMPK activity still live longer than WT worms¹¹, because many other targets of IIS keep functioning. Consistently, it has been a puzzle why defective sensory perception extends lifespan through suppressing IIS but suppresses the longevity of *daf-2(-)* mutants¹¹. Our findings suggest that this is because sensory defects suppress AMPK and abolish part of the longevity effect of *daf-2(-)*.

5) The authors demonstrated that enhancing sensory cilia functions via overexpressing *daf-19* upregulated AMPK activity by showing increased p-AMPK level. Considering that *daf-19* was downregulated during aging, and sensory neural functions were also deteriorated, as the authors depicted in the Fig. 6h, one can expect that p-AMPK level is decreased during aging. However, Fig. 4b, 4e, and S4c all showed that p-AMPK level increased in old worms compared to young worms, and no description about this discrepancy is written in the manuscript. The authors should describe the age-dependent change in p-AMPK level. Otherwise it does not explain the authors’ suggested model.

-- Sorry for your confusion. AMPK (AMP activated protein kinase), by its name, is known to be activated by a high AMP:ATP ratio¹². It is reported that the AMP:ATP ratio increases in aged worms, potentially due to the reduced food intake¹³⁻¹⁵. Therefore, p-AMPK level increases in old worms. We have clarified this issue in the revised manuscript.

However, this upregulation of p-AMPK is not enough to maintain the metabolic homeostasis, because increasing AMPK activity in wildtype worms is known to promote the metabolic homeostasis and longevity^{13,16,17}. Our findings indicate that the decrease of IFT and sensory perception is responsible for the insufficient AMPK activity in aged worms.

6) In Fig. 2d, they need to show the data and the quantification for proteins in cilia, as in Fig. 2a to support their claim.

-- We have performed the quantification for proteins in cilia as suggested. Please see Figure 2d in the revised manuscript.

7) The authors should provide the raw data of all Western blot they present in this study in supplementary materials with proper size markers labeled.

-- We have provided the requested data in Supplementary Figure 8.

Minor concerns

1) The authors used day 10 worms of lifespan assays for IFT velocity and used them again for lifespan assay after recovery in Fig. 6d. Does the exposure to levamisole at day 10 affect the result of lifespan assay?

-- A short answer is “no”. We tried to recover the worms as fast as possible in our experiments. As showed in Supplementary Figure 7e and Supplementary Table 3, the maximal lifespan and median lifespan and the survival curve of total worms after exposure to levamisole are similar with the WT worms under normal conditions. So, in our study, the exposure to levamisole at day 10 have little effect on the result of lifespan assay. We have further clarified this issue in the revised manuscript.

2) It would be better if authors guide readers to look into supplementary table S3 for the number of independently repeated experiments, total number of worms counted, and the statistical analysis of the presented data in the figure legends.

-- Thanks for your suggestion! The figure legends have been modified as suggested.

3) On page 8, the authors described that “a mild *daf-19* RNAi did not disrupt ciliogenesis but abolished the elevated motilities of OSM-3::GFP and OSM-6::GFP in *daf-2(-)* mutants at D10 to WT levels and substantially downregulated the velocity of CHE-3::GFP and CHE-11::GFP in *daf-2(-)* mutants (Supplementary Fig. 2d-2f).” However, in Fig. S2e, *daf-19* RNAi did not reduce the motility of OMS-3::GFP or OSM-6::GFP in *daf-2(-)* mutants at day 10, as there are no significant changes between *daf-2(-)* *luc2* RNAi and *daf-2(-)* *daf-19* RNAi animals. The authors should revise the manuscript to provide proper explanation of the data.

-- By motility, we mean both velocity and frequency. Whereas frequency in Supplementary Figure 3e (previously Supplementary Figure 2e) unchanged, the velocity in Supplementary Figure 3d (previously Supplementary Figure 2f) decreased upon *daf-19* RNAi. We have rephrased the corresponding statement to avoid further misunderstanding.

4) The authors should provide detailed explanation for the criteria of quantifying CRTC-1::RFP subcellular localization shown in Fig. 4c.

-- Thanks a lot for your suggestion! Please see “Figure 2 for reviewers” for our criteria. In the revised manuscript, we have used a more accurate way to quantify CRTC-1::RFP subcellular localization in the revised Figure 4d (Figure 4c in the previous manuscript), by their fluorescent intensity but not by three categories.

Figure 2 for reviewer. The criteria for quantifying CRTC-1::RFP subcellular localization. Bar: 10 μ m.

5) What does the percentage mean in the Fig. 6d? The authors should provide detailed explanation of the criteria for dividing fast and slow IFT-displaying worm groups.

-- Thanks a lot for your comment! In this assay, we ranked worms by their velocity from the fastest to the slowest. Fast 20% means the top 20% worms in this rank and slow 20% are the bottom 20% worms in this rank. We have detailed the criteria for fast and slow IFT-displaying worm groups in the revised manuscript.

6) There is no description for Fig. S2f in the figure legends.

-- Sorry for the mistake. The mistake has been corrected in the revised manuscript.

7) On page 9, The authors described, “IFT mutants of *osm-3(-)* and *osm-6(-)* exhibited

severe cilia defects and IIS target genes were regulated as reported at D1 (Supplementary Fig. 3a). In aged worms, IIS exhibited a decreased modulation by sensory perception, as multiple IIS target genes were no longer changed upon mutating *osm-3* or *osm-6* at D10 (Supplementary Fig. 3a).” I think the authors made these sentences very vague (perhaps intentionally) because *osm-3(-)* and *osm-6(-)* actually UPregulate DAF-16 target genes, which contribute to longevity as many previous papers reported. However, I think this will make many readers even more confused, and I suggest that the authors revise the sentences to incorporate the direction of gene expression changes. If necessary, the authors may discuss this confusing issue in the discussion further.

-- Thanks for your comment! For Supplementary Figure 4a (previously Supplementary Figure 3a), we have further clarified its description with clear direction of gene expression changes in the revised manuscript. We have also further discussed this seemingly “confusing” issue.

8) Have the authors used neuron-specific RNAi strain backgrounds (TU3410 or SYD0779) for lifespan assays? The authors should then provide detailed strain information in the supplementary table 3, lifespan analysis tab to provide precise information regarding strains they used.

-- Worms with TU3401 background were used for lifespan assays. Strain information is now included in Supplementary Table 3 – Lifespan Analysis tab. Thank you for your suggestion!

References

- 1 Calixto, A., Chelur, D., Topalidou, I., Chen, X. & Chalfie, M. Enhanced neuronal RNAi in *C. elegans* using SID-1. *Nat Methods* **7**, 554-559, doi:10.1038/nmeth.1463 (2010).
- 2 Fujiwara, M., Aoyama, I., Hino, T., Teramoto, T. & Ishihara, T. Gonadal Maturation Changes

- Chemotaxis Behavior and Neural Processing in the Olfactory Circuit of *Caenorhabditis elegans*. *Curr Biol* **26**, 1522-1531, doi:10.1016/j.cub.2016.04.058 (2016).
- 3 Broekhuis, J. R., Leong, W. Y. & Jansen, G. Regulation of cilium length and intraflagellar transport. *Int Rev Cell Mol Biol* **303**, 101-138, doi:10.1016/B978-0-12-407697-6.00003-9 (2013).
- 4 Ishikawa, H. & Marshall, W. F. Intraflagellar Transport and Ciliary Dynamics. *Cold Spring Harbor perspectives in biology* **9**, doi:10.1101/cshperspect.a021998 (2017).
- 5 Boehlke, C. *et al.* Primary cilia regulate mTORC1 activity and cell size through Lkb1. *Nature cell biology* **12**, 1115-1122, doi:10.1038/ncb2117 (2010).
- 6 Mick, D. U. *et al.* Proteomics of Primary Cilia by Proximity Labeling. *Developmental cell* **35**, 497-512, doi:10.1016/j.devcel.2015.10.015 (2015).
- 7 Zhong, M. *et al.* Tumor Suppressor Folliculin Regulates mTORC1 through Primary Cilia. *The Journal of biological chemistry* **291**, 11689-11697, doi:10.1074/jbc.M116.719997 (2016).
- 8 Kenyon, C. J. The genetics of ageing. *Nature* **464**, 504-512, doi:10.1038/nature08980 (2010).
- 9 Murphy, C. T. *et al.* Genes that act downstream of DAF-16 to influence the lifespan of *Caenorhabditis elegans*. *Nature* **424**, 277-283, doi:10.1038/nature01789 (2003).
- 10 Bunu, G. *et al.* SynergyAge, a curated database for synergistic and antagonistic interactions of longevity-associated genes. *Sci Data* **7**, 366, doi:10.1038/s41597-020-00710-z (2020).
- 11 Apfeld, J. & Kenyon, C. Regulation of lifespan by sensory perception in *Caenorhabditis elegans*. *Nature* **402**, 804-809, doi:10.1038/45544 (1999).
- 12 Herzig, S. & Shaw, R. J. AMPK: guardian of metabolism and mitochondrial homeostasis. *Nature reviews. Molecular cell biology* **19**, 121-135, doi:10.1038/nrm.2017.95 (2018).
- 13 Apfeld, J., O'Connor, G., McDonagh, T., DiStefano, P. S. & Curtis, R. The AMP-activated protein kinase AAK-2 links energy levels and insulin-like signals to lifespan in *C. elegans*. *Genes & development* **18**, 3004-3009, doi:10.1101/gad.1255404 (2004).
- 14 Xu, Y., He, Z., Song, M., Zhou, Y. & Shen, Y. A microRNA switch controls dietary restriction-induced longevity through Wnt signaling. *EMBO reports*, doi:10.15252/embr.201846888 (2019).
- 15 Huang, C., Xiong, C. & Kornfeld, K. Measurements of age-related changes of physiological processes that predict lifespan of *Caenorhabditis elegans*. *Proceedings of the National Academy of Sciences of the United States of America* **101**, 8084-8089, doi:10.1073/pnas.0400848101 (2004).
- 16 Burkewitz, K. *et al.* Neuronal CRT-1 governs systemic mitochondrial metabolism and lifespan via a catecholamine signal. *Cell* **160**, 842-855, doi:10.1016/j.cell.2015.02.004 (2015).
- 17 Mair, W. *et al.* Lifespan extension induced by AMPK and calcineurin is mediated by CRT-1 and CREB. *Nature* **470**, 404-408, doi:10.1038/nature09706 (2011).

Reviewers' Comments:

Reviewer #1:

Remarks to the Author:

I am fine with the revision and support its publication.

Reviewer #2:

Remarks to the Author:

The manuscript by Zhang et al has been revised, including changes to text, figures, and additional experiments. I have looked at my suggested revisions and found them to be satisfactory. I also checked other reviewers' suggested changes, and find that the authors have been dutiful in revising their manuscript. In particular, the cell-specific RNAi of daf-19 represent a nice, complementary study to assess the role of this transcription factor during ageing. The Discussion is also substantially revised.

I find that the manuscript is satisfactorily revised, and ready for publication.

Reviewer #3:

Remarks to the Author:

The authors addressed most of my comments sufficiently. I have one minor issue that did not seem to be addressed in the revised manuscript. For the minor concerns 4), I cannot find the criteria for quantifying CRTC-1::RFP subcellular localization in the manuscript. It will be better if the detailed information is written in the Methods.

Our detailed responses to the reviewers' concerns are listed below on a point-by-point basis.

Reviewer #1 (Remarks to the Author):

I am fine with the revision and support its publication.

Reviewer #2 (Remarks to the Author):

The manuscript by Zhang et al has been revised, including changes to text, figures, and additional experiments. I have looked at my suggested revisions and found them to be satisfactory. I also checked other reviewers' suggested changes, and find that the authors have been dutiful in revising their manuscript. In particular, the cell-specific RNAi of daf-19 represent a nice, complementary study to assess the role of this transcription factor during ageing. The Discussion is also substantially revised.

I find that the manuscript is satisfactorily revised, and ready for publication.

Reviewer #3 (Remarks to the Author):

The authors addressed most of my comments sufficiently. I have one minor issue that did not seem to be addressed in the revised manuscript. For the minor concerns 4), I cannot find the criteria for quantifying CRTC-1::RFP subcellular localization in the manuscript. It will be better if the detailed information is written in the Methods.

-- Sorry for our oversimplified explanation in the previous rebuttal letter and the manuscript. In the very first version of our manuscript, we arbitrarily categorized the subcellular localization of CRTC-1::RFP into three types roughly by the nuclear RFP signal. The representative images of each type and the corresponding quantification

are shown in Figure 1 for the reviewer (Figure 2 for the reviewer in the previous rebuttal letter).

In the revised manuscript, we took a more accurate quantification. We measured the mean fluorescent intensity of CRTC-1::RFP in the nucleus and cytoplasm of the intestinal cells and thereby calculated its nucleo-cytoplasmic ratio (Figure 4d). The corresponding description has been included in the “Methods” section, as suggested.

Figure 1 for the reviewer. The criteria for quantifying CRTC-1::RFP subcellular localization. Bar: 10 μ m.

Finally, we would like to thank all the reviewers again for reviewing this manuscript and your constructive comments!